# Endothelial tip-cell position, filopodia formation and biomechanics require BMPR2 expression and signaling
Christian Hiepen [1,10,11] ✉, Mounir Benamar [1,11], Jorge Barrasa-Fano [2], Mar Condor[2], Mustafa Ilhan[1,3], Juliane Münch [4], Nurcan Hastar[1], Yannic Kerkhoff [1], Gregory S. Harms[5], Thorsten Mielke[6], Benjamin Koenig[1,7], Stephan Block[1], Oliver Rocks[8], Salim Abdelilah-Seyfried [4], Hans Van Oosterwyck [2,9] & Petra Knaus [1] ✉

Blood vessel formation relies on biochemical and mechanical signals, particularly during sprouting angiogenesis when endothelial tip cells (TCs) guide sprouting through filopodia formation. The contribution of BMP receptors in defining tip-cell characteristics is poorly understood. Our study combines genetic, biochemical, and molecular methods together with 3D traction force microscopy, which reveals an essential role of BMPR2 for actin-driven filopodia formation and mechanical properties of endothelial cells (ECs). Targeting of Bmpr2 reduced sprouting angiogenesis in zebrafish and BMPR2-deficient human ECs formed fewer filopodia, affecting cell migration and actomyosin localization. Spheroid assays revealed a reduced sprouting of BMPR2-deficient ECs in fibrin gels. Even more strikingly, in mosaic spheroids, BMPR2-deficient ECs failed to acquire tip-cell positions. Yet, 3D traction force microscopy revealed that these distinct cell behaviors of BMPR2-deficient tip cells cannot be explained by differences in force-induced matrix deformations, even though these cells adopted distinct cone-shaped morphologies. Notably, BMPR2 positively regulates local CDC42 activity at the plasma membrane to promote filopodia formation. Our findings reveal that BMPR2 functions as a nexus integrating biochemical and biomechanical processes crucial for TCs during angiogenesis.

The vasculature consists of a branched network of blood vessels that support proper functions of all other organs. It is one of the first higher ordered tissue structures to develop during embryogenesis. Blood vessels are comprised of endothelial cells (ECs) which line their inside and are associated to mural cells (MCs) which form the outer layers. New blood vessels are formed by sprouting from pre-existing ones during a process known as sprouting angiogenesis[1]. Following the activation of the endothelium by pro-angiogenic growth factors, ECs can specify into tip cells (TC) which are highly invasive and migratory during angiogenic sprouting[2]. Neighboring

ECs specify into stalk-cells (SC) via tip cell induced lateral signaling which promotes stalk cell-specific gene expression. This creates a feedback loop that reinforces TC vs. SC competence within the sprout. SCs follow the TC, ensuring connectivity of the new sprout to the original blood vessel and supporting its outgrowth by proliferation. Both, developmental[3–8] and tumor vascularization studies[9,10] strongly suggest that BMP signaling is critically involved in sprouting angiogenesis[11]. Yet, we still lack a detailed understanding of the molecular machinery involved in integrating biochemical and biomechanical signals during sprouting angiogenesis.

[1]Freie Universität Berlin, Institute for Chemistry and Biochemistry, Thielallee 63, 14195 Berlin, Germany. [2]KU Leuven, Department of Mechanical Engineering, Biomechanics section, Leuven, Celestijnenlaan 300 C, 3001 Leuven, Belgium. [3]Berlin School of Integrative Oncology, Augustenburger Platz 1, D-13353 Berlin, Germany. [4]Universität Potsdam, Institute of Biochemistry and Biology, Karl-Liebknecht Strasse 24-25, 14476 Potsdam-Golm, Germany. [5]Universitätsmedizin, Johannes Gutenberg-Universität Mainz, Cell Biology Unit, Imaging Core Facility and the Research Center for Immune Intervention, Langenbeckstraße 1, 55131 Mainz, Germany. [6]Max-Planck-Institute for Molecular Genetics, Microscopy & Cryo-Electron Microscopy, Ihnestraße 63-73, 14195 Berlin, Germany. [7]Leibniz Forschungsinstitut für Molekulare Pharmakologie (FMP), Berlin, Germany. [8]Charité - Universitätsmedizin Berlin, Systemic Cell Dynamics, Charitéplatz 1, 10117 Berlin, Germany. [9]KU Leuven, Prometheus Division of Skeletal Tissue Engineering, Leuven, Belgium. [10]Present address: Westphalian University of Applied Sciences, August-Schmidt-Ring 10, 45665 Recklinghausen, Germany. [11]These authors contributed equally: Christian Hiepen, Mounir Benamar. ✉e-mail: christian.hiepen@w-hs.de; Petra.knaus@fu-berlin.de

Bone Morphogenetic Proteins (BMPs) belong to the transforming growth factor beta (TGF-β) superfamily and fulfill important functions in the vasculature[12,13]. BMPs bind to hetero-oligomeric receptor complexes comprising a set of type I and type II serine/threonine kinases[14]. Circulating BMP-9/-10 in human plasma maintains EC quiescence[15] by inducing suppressor of mothers against decapentaplegic (SMAD)-1/5/9 transcription factor[16] signaling. This canonical SMAD signaling maintains vascular homeostasis and emerges from the type I receptor activin receptor-like kinase 1 (ALK1) which is highly abundant at the surface of ECs. The constitutively- active type II receptor BMPR2 is required for ligand-induced trans-activation of ALK1[17]. Beyond this role, BMPR2 is involved in signaling events that do not rely on signal transduction via transcription factor SMADs. Such signaling that is mechanistically different to BMPR-dependent SMAD activation is often referred to as "non- canonical" BMP signaling or "non-SMAD" signaling[18] and is involved in e.g. regulating cytoskeletal organization[19]. BMPR2 has a long cytosolic tail, which may function as a scaffold for signaling molecules that diversify signaling outcomes[20,21]. Those include common effectors of receptor tyrosine kinases such as phosphoinositide 3-kinases (PI3K)[21] and c-Src[22]. However, in ECs, particularly during their angiogenic activation, little is known about BMPR2-specific functions and potential roles of this specific BMP type II receptor in non- canonical signaling pathways related to sprouting angiogenesis.

Signaling by BMP-ligands including BMP-2/-4/-6/-7 activates ECs of various vascular beds while BMP-9/-10 circulates in the plasma acting as quiescence factors[23–28]. However, ECs are activated by angiogenic cues from outside the vessel. Pro-angiogenic BMPs were proposed to act as soluble gradients and/or bound to the extracellular matrix (ECM)[29]. Recently, we demonstrated that soluble gradients of BMP-2 and BMP-6 are sufficient to induce chemotaxis of human umbilical vein endothelial cells (HUVECs) in vitro[30]. In-vivo, such BMP-dependent EC activation may promote sprouting angiogenesis[31,32]. Current advancements in 3D culturing techniques, such as the use of spheroids, now allow for in vitro recapitulation of key aspects of this complex process[33].

During sprouting angiogenesis, single TCs residing at the tips of nascent sprouts lead sprout outgrowth by migration. They utilize filamentous (F-) actin-driven filopodia to probe and interact with the ECM. Filopodia formation depends on growth factor-induced signaling mechanisms that facilitate actin polymerization and focal adhesion (FA) assembly. Hence, TC migration involves contractile actomyosin reorganization linked to FAs and integrins[34]. In emerging sprouts, TCs mainly migrate and pull on the ECM, while the following SCs proliferate to support sprout elongation. Pulling occurs when TC contractility allows integrins to bind ligands, such as the ECM protein Fibrin[35–38]. Interference with the Rho/ROCK pathway is sufficient for relaxation of TC contractile actomyosin and for the release of pulling forces towards the ECM (supplementary Movie 1). Although not absolutely required for sprouting[39], filopodia seem to allow for a more efficient sprouting process[40–42].

Previous research by others revealed a role for the Rho GTPase Cell Division Cycle 42 (CDC42) in BMP-induced sprouting angiogenesis at the zebrafish caudal vein plexus (CVP)[43]. CDC42 is a small GTPase spatially controlled at the plasma membrane involved in regulating actin polymerization, filopodia formation and FAs assembly[44]. CDC42 deletion in mouse retina led to filopodia deficiency, reduced sprouting angiogenesis, loss of endothelial polarity and impaired EC adhesion[44]. Loss of CDC42 is linked to the development of capillary-venous malformations[45,46]. Although several BMP ligands are recognized as activators of Rho GTPase signaling in various cell types[19,47,48], the specific mechanisms and interdependencies connecting the abundance of particular BMPRs on the cell surface to the activation of CDC42 in endothelial cells are not yet fully understood.

In this study, we have unveiled the pivotal role played by BMPR2 in regulating CDC42-dependent filopodia formation and the associated biomechanical processes during angiogenic sprouting. Targeting Bmpr2 using CRISPR/Cas9 or antisense morpholino oligos in zebrafish greatly reduced the number of vascular sprouts in the emerging vascular network of the caudal vein plexus. In human BMPR2-deficient ECs, the number of filopodia at the polarized leading edge was reduced. Combining cell biological and biochemical studies revealed that BMPR2 affects focal adhesion morphology and actomyosin contractility particularly at the leading edge of migrating cells. Consequently, we posit that BMPR2 exerts precise spatial control over important biomechanical properties transduced via the acto-myosin network of ECs. Functionally, we found that BMPR2 protein levels are decisive for the mode of EC migration in 2D. BMPR2 deficiency and filopodia loss exacerbated collective EC migration to a degree that impeded an efficient forward movement and altered subcellular actomyosin localization. Employing 3D-sprouting assays enabled us to determine the precise localization of individual TCs and SCs within sprouts. This approach revealed that BMPR2 is indispensable for ECs acquiring the TC position in spheroids and for their efficient sprouting. While there were no major differences in average cell-induced matrix deformations between BMPR2^{wt} and BMPR2^{+/-} ECs, the lack of filopodia in BMPR2^{+/-} TCs resulted in a cone-shaped TC morphology leading to altered TC mechanical properties. Finally, we found that BMPR2 is an essential driver of CDC42 activity at the endothelial plasma membrane by acting both on activating CDC42 and downstream by regulating e.g. the filopodial localization of CDC42 effectors. This positions BMPR2 as a key integrator of biochemical and biomechanical signal, essential for coordinating critical endothelial functions like filopodia formation and cell migration during sprouting angiogenesis.

## Results
### BMPR2 is required for zebrafish CVP sprouting angiogenesis and human EC filopodia formation

Sprouting angiogenesis during formation of the zebrafish caudal vein is dependent on BMP and independent of VEGF-A signaling[45]. During embryogenesis, zebrafish express two BMP type II receptors Bmpr2a and Bmpr2b that could mediate this effect[45,46]. To assess this possibility, we depleted *Bmpr2b* by injections of an antisense morpholino oligo (MO) targeting a splice site into the vascular reporter transgenic line *Tg(kdrl:GFP)^{s843}*, which substantially reduced *bmpr2b* spliced transcript levels at 8 h post fertilization (hpf) (Supplementary Fig. 1). In addition, we applied a crispant-approach[47] where mutations were introduced into exon 1 and exon 3 of the *bmpr2b* gene (Supplementary Fig. 2). This was confirmed by the presence of overlapping peaks upon Sanger-Sequencing of PCR products of corresponding regions in 93% (exon 1) and 100% (exon 3) of embryos (Supplementary Fig. 2). By 25 hpf, *bmpr2b* morphants had significantly fewer sprouts protruding from the CVP, a phenotype that was recapitulated in 26hpf *bmpr2b*-crispants (Fig. 1a, b). This resulted in a deficient plexus in *bmpr2b*-morphants and crispants at 32 hpf indicated by a lower number of fenestrations in the plexus and a reduced plexus size (Supplementary Fig. 3, 4), This phenotype was not simply caused by a developmental delay since other hallmarks of development were not affected. These included the arterial intersegmental vessels (aISV´s), which were properly formed in control and *bmpr2b*-crispant embryos (Supplementary Fig. 5). Furthermore, by using the tip cell marker CD34, we could confirm the presence of CD34-positive tip cells protruding from the CVP of control zebrafish compared to a reduced amount of tip cells in *bmpr2b*-crispants (Fig. 1c). Hence, BMP signaling via Bmpr2b regulates zebrafish CVP (but not arterial) sprouting, adding importance for Bmpr2 to the known BMP-dependent mechanism for vascular plexus formation in zebrafish[43,45].

To next address whether BMPR2 deficiency impairs the migratory phenotype of human ECs, we took advantage of a previously established and extensively characterized immortalized human umbilical vein EC (HUVEC) cell-line depleted for BMPR2 expression by CRISPR/Cas9 (BMPR2^{+/-}), resulting in BMPR2 haploinsufficiency[48,49] (Supplementary Fig. 6). To complement the EC line, we used primary HUVECs, treated with self-delivering siRNA to efficiently knock down (KD) BMPR2 (Supplementary Fig. 6). To mimic typical TC-related characteristics in 2D in vitro (i.e. polarization, filopodia formation and migration), we used seeding inserts. Upon removal, they leave behind a cell-free gap towards which ECs

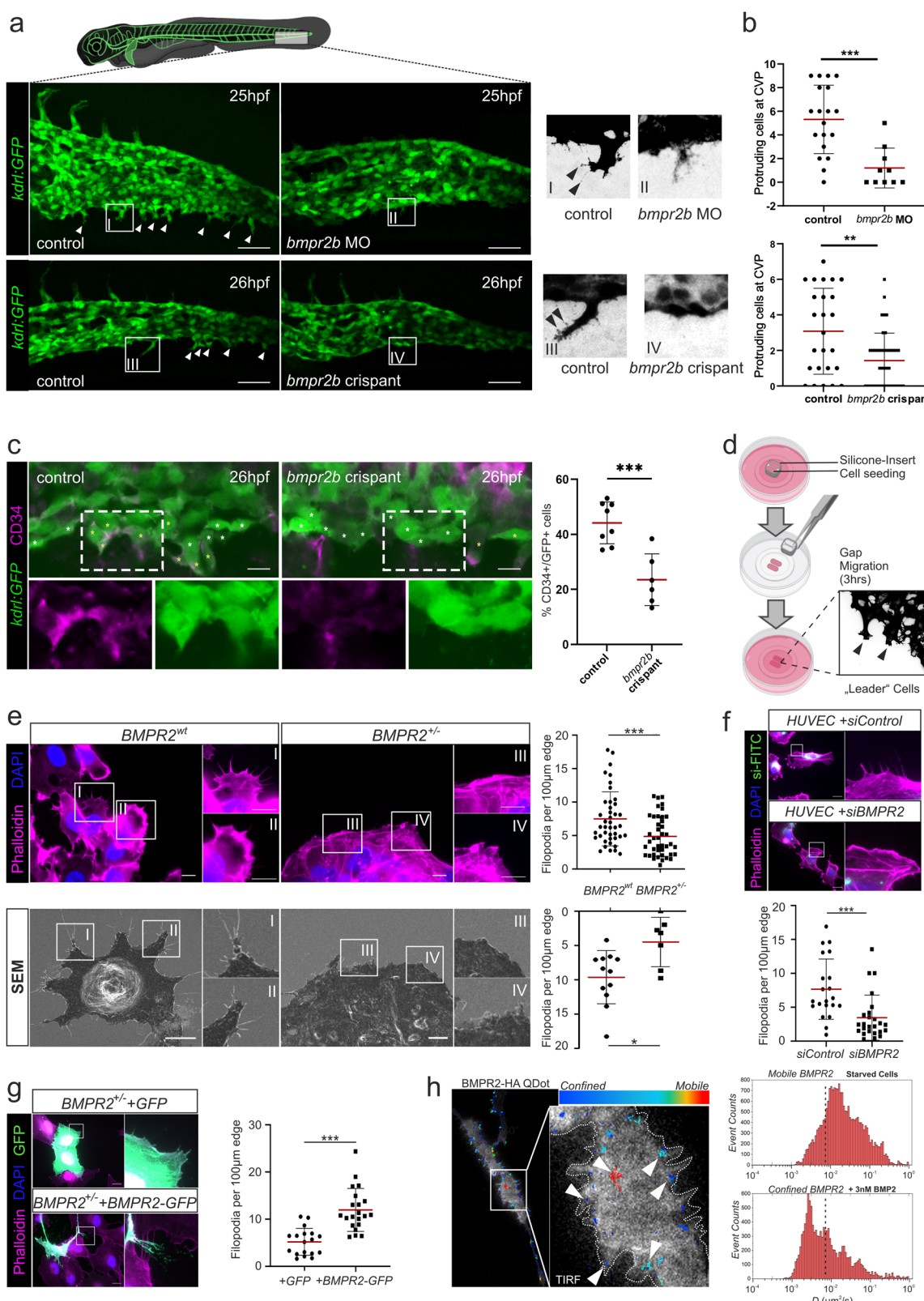

can migrate. For this, ECs require to polarize and form a leading edge in the presence of pro-angiogenic cues. In our approach, those are delivered by the EC-activation medium (Fig. 1d) by polarizing and forming a leading edge in the presence of pro-angiogenic cues delivered by the EC-activation medium (Fig. 1d). In this approach, although ECs cannot fully specialize as TCs due to the lack of a proper 3D environment, the first row of "leader cells" adopts

TC-like characteristics when it comes to the early steps of migration, notably recapitulating filopodia formation at the leading edge. F-actin staining of leader cells revealed that BMPR2$^{+/-}$ ECs display significantly less filopodia per cell edge when compared to their wild-type (WT) counterparts (Fig. 1e, upper). This finding was observed also at higher magnification by Scanning Electron Microscopy (SEM) (Fig. 1e, lower). In line with this, siRNA-

**Fig. 1 | BMPR2 is required for CVP sprouting angiogenesis and EC filopodia formation, contributing to the endothelial tip cell phenotype. a** Zebrafish embryo expressing the endothelial Tg(kdrl-GFP) reporter transgene and maximum projection images of the caudal vein plexus (CVP) at 25 h post fertilization (hpf) from control or *bmpr2*b MO-treated embryos (upper) and control or *bmpr2*b crispant-treated embryos (lower) (arrows indicate individual tip cells). Scale bar: 50 µm. Right, close-up of regions (I and II) of interest depicted in *bmpr2*b MO-treated embryos (upper) and close-up of regions (III and IV) of interest depicted in *bmpr2*b crispant-treated embryos (lower) (arrows indicate individual filopodia at tip cell). **b** Quantification of the number of protruding cells at the CVP of control or *bmpr2*b MO-treated zebrafish and control or *bmpr2*b crispant-treated zebrafish at 25hpf. ***$p < 0.001$. **c** Representative images of optical sections showing part of the caudal vein plexus (CVP) in Tg(kdrl:GFP) endothelial reporter embryos. In control embryos, CD34 is present in many cells along the front of the CVP (yellow asterisks) whereas in *bmpr2*b-crispants most of the cells lack CD34 (indicated by white asterisks). Scale bar: 10 µm. Quantification of CD34$^+$/ GFP$^+$ cells in comparison to GFP$^+$ cells in the CVP from Tg(kdrl:GFP) endothelial reporter embryos. ***$p < 0.005$. **d** Scheme describing formation of leader cells by gap closure assay using silicone-inserts. Cells are seeded in a silicone-insert on a glass coverslip. After cell-adherence, the silicone-insert is removed creating a cell-free gap towards which the first rows of cells polarize and migrate. Samples are then fixed and treated for immunofluorescence staining. **e** Immunofluorescence pictures of control and BMPR2$^{+/-}$ ECs, Phalloidin (magenta), DAPI (blue) (top). Scanning-Electron

Micrographs (SEM) of control and BMPR2$^{+/-}$ ECs (bottom). Inlets show regions of interest. Respective quantifications of the number of filopodia per 100 µm of cell edge for immunofluorescence (upper) and SEM (lower) analysis are shown on the right. *$p < 0.05$; ***$p < 0.001$. Scale bar: 10 µm. **f** Immunofluorescence staining of HUVECs transfected with FITC-labeled control siRNA (siControl) or a 1:1 mix of FITC-labeled control siRNA and BMPR2-targeting siRNA (siBMPR2). Phalloidin (magenta), DAPI (blue), FITC (green). Inlets show regions of interest. Quantification of the number of filopodia per 100 µm of cell edge is shown below. ***$p < 0.001$. Scale bar: 20 µm. **g** Immunofluorescence staining of BMPR2$^{+/-}$ ECs overexpressing GFP control (upper) or BMPR2-GFP for filopodia rescue (lower). Phalloidin (magenta), DAPI (blue), GFP (green). Insets show regions of interest. Quantification of the number of filopodia per 100 µm of cell edge is shown on the right. ***$p < 0.001$. Scale bar: 20 µm. **h** Single particle trajectories of individual BMPR2 molecules. Trajectories were recorded by Total Internal Reflection Fluorescence (TIRF) microscopy. For this, BMPR2-HA was overexpressed in Cos7 cells and labeled with individual Quantum-Dots (Qdot). Trajectories were recorded for 25 s and analyzed via tracking software. Trajectories were overlaid onto F-Actin signal (upon co-transfection of LifeAct-GFP) and color coded relative to their magnitude (immobile BMPR2: Blue; mobile BMPR2: Red). Arrowheads indicate different BMPR2 mobilities across different cellular sub-compartments (left). BMP2 induced confinement of BMPR2 (right): BMPR2 trajectories in Cos7 cells before (upper) and after stimulation with 3 nM BMP2 (lower). Reduction in diffusivity of individual BMPR2 molecules in µm$^2$/sec is shown after stimulation with BMP2.

mediated KD of BMPR2 in HUVECs also resulted in a significantly reduced number of filopodia or even a filopodia-free (blunted) morphology at the leading edge (Fig. 1f). However, in some cells, a few isolated filopodia remained upon BMPR2 depletion. High throughput analysis using FiloQuant[50] did not detect significant differences in the length of the remaining filopodia neither in siBMPR2-treated HUVECs nor in BMPR2$^{+/-}$ cells when compared to controls (Supplementary Fig. 7). Together this shows, that immortalized ECs and primary HUVECs both depend on BMPR2 for filopodia formation of leader cells as an early step in polarized migration in 2D. We next aimed to rescue the filopodial phenotype of BMPR2$^{+/-}$ ECs by re-expressing BMPR2 as a GFP-tagged fusion protein. BMPR2-GFP overexpression effectively rescued filopodia deficiency in BMPR2$^{+/-}$ ECs when compared to controls transfected (GFP only) (Fig. 1g). Moreover, excessive filopodia formation was observed when BMPR2-GFP was ectopically expressed in primary HUVECs (Supplementary Fig. 8). We furthermore found BMPR2 foci along filopodia but also at their tips and close to sites where filopodia originate (herein referred to as filopodia base) (Supplementary Fig. 8). The same result was observed and quantified for ectopic BMPR2-GFP expression in the parental EC line (BMPR2$^{wt}$) (Supplementary Fig. 9). This shows that in both EC model systems, ectopic expression of BMPR2 positively correlates with greater number of filopodia. Interestingly, the transfection of different amounts of BMPR2-GFP plasmid positively correlated with increased appearance of so called "migration tracks" (aka. cellular remnants). This phenomenon of cells migrating in vitro occurs during rear-end retraction and indicates strong adherence of cells to their substratum due to enrichment in FAs or disturbed FA disassembly[51]. Furthermore, quantitative analysis of filopodia formation in cells transfected with a set amount of BMPR2-GFP plasmid revealed a positive correlation between GFP signal intensity and filopodia number, further underlining a dose-dependency for BMPR2 expression and enhanced filopodia formation in ECs (Supplementary Fig. 10). To investigate the role of BMP-induced SMAD signaling in BMPR2-mediated EC filopodia formation, we overexpressed BMPR2 under concomitant inhibition of pan type I receptor kinase activity, required for SMAD activation, by using the small molecule inhibitor LDN-193189 (LDN)[52]. BMP type I receptor inhibition led to strong reduction in basal SMAD phosphorylation in EA.hy926 and HUVEC when cells were cultured in EC activation media supplemented with LDN-193189 (Supplementary Fig. 11). BMPR2-overexpressing ECs showed significantly increased filopodia formation even in the presence of LDN (Supplementary Fig. 12). These results were recapitulated in ECs overexpressing a Short Form (SF) of BMPR2 lacking its intracellular tail (Supplementary Fig. 13). The BMPR2-SF represents a rare

natural variant of BMPR2[53]. Conversely, BMPR2$^{+/-}$ ECs overexpressing a BMPR2 variant with inactive kinase (BMPR2-K230R) showed reduced filopodia numbers compared to ECs expressing a functional BMPR2 (BMPR2-GFP) (Supplementary Fig. 14). Finally, we assessed the role of specific type 1 receptors and co-receptors in filopodia formation by overexpressing Alk1, Alk2, Alk3 and Endoglin in BMPR2$^{+/-}$ ECs. Neither type 1 receptors nor Endoglin overexpression were able to rescue filopodia formation in BMPR2-depleted cells (Supplementary Fig. 15). These results suggest together that BMPR2-dependent filopodia formation does not directly rely on type I receptor signaling or on the BMPR2 intracellular cytosolic tail, but does require BMPR2 kinase activity.

Our previous investigation on BMPR2 subcellular localization (Fig. 1g) suggested its membrane presence along all regions of the filopodium. We next investigated further on BMPR2 subcellular localization, particularly its membrane mobility in proximity to cell edges of living cells. Membrane mobility of distinct BMPRs can report on their respective involvement in SMAD vs. non-SMAD signaling complexes[54]. For this we performed single particle tracking microscopy (SPTM) using quantum-dot (QDot)-labeled HA-tagged BMPR2 in living cells[54,55]. After successful validation of BMPR2 labeling efficacy (Supplementary Fig. 16), SPTM analysis for single QDot-labeled BMPR2 molecules (Fig. 1h, left) revealed several BMPR2 foci at the cellular periphery (F-actin labeled by LifeActGFP), close to filopodia base. Confinement of mobile BMPR2 upon addition of ligand indicates BMPR2 in complex with BMP and BMP type I receptors, forming active signaling complexes termed BMP-induced signaling complexes (BISCs)[54,56]. We could previously show, that BISC activation is associated with non-canonical signaling responses[54]. To prove that the observed mobility behavior of BMPR2 at filopodial bases is indeed indicative for such BISC formation and in line with our previously published data[54], we induced confinement of mobile, i.e. ligand-unbound BMPR2 receptors in starved cells, upon pulsed stimulation with BMP-2. BMP2 stimulation led to a reduction of BMPR2 lateral mobility at the filopodia base as shown by shift of individual receptor displacements (Fig. 1h, right). This result is indicative for a confinement of BMPR2 induced by BMP2 and enrollment of BMPR2 into BISC signaling complexes at those sites. Together these data show that BMPR2 is located at all filopodia regions, although with different lateral mobilities, revealing a particular involvement of BMPR2 in BMPR signaling complexes at the filopodia base related to non-canonical signaling.

In sum, these data show that BMPR2 expression levels correlate with ECs ability to form filopodia at their leading edge using two independent cellular and one animal model system. Furthermore, BMPR2 is present throughout the filopodial membrane but adopts a primarily confined lateral

mobility at the base of the filopodia upon BMP stimulation, indicative for BISC formation and non-canonical signaling emanating at those sites.

## Loss of BMPR2 alters EC migration dynamics, tube-formation and focal adhesion morphology accompanied by defects in spatial actomyosin organization

While BMP-2 and other BMP family members facilitate EC motility[13], their role as classical angiogenic guidance cues is still under debate, despite data showing chemotactic responses of ECs towards BMP gradients[24,30]. An interesting hypothesis is that instead of directly inducing EC polarity and migration like typical endothelial guidance cues, BMPs promote only the migratory velocity of cells that already display an established front-to-rear end polarity. To investigate on this further, we performed dynamic analysis of gap closure migration assays under EC activating medium conditions. We found that BMPR2$^{+/-}$ cells migrate with reduced efficacy into the cell-free gap when compared to controls (Fig. 2a). However, more revealing was dynamic cell migration component analysis on image stacks of wildtype (green) and BMPR2$^{+/-}$ (magenta) ECs simultaneously migrating over 16 h within the same gap-field-of-view (Fig. 2b, upper left). Particle image velocimetry analysis applied to these data (Fig. 2b, lower left) shows that movements of BMPR2$^{+/-}$ cells appear in larger clusters, with individual cells within the cluster adopting the same directionality, creating a swirl-like displacement pattern (Fig. 2b, inlet I). On the other hand, BMPR2$^{wt}$ cells appeared to form only small clusters and displayed lesser group dependency in their directionality of movement (Fig. 2b, inlet II). Subsequent migration component analysis, in which all trajectories were decomposed into directional (velocity) and random (diffusivity) contributions to cell migration[57], confirmed reduced migratory diffusivity of BMPR2$^{+/-}$ ECs with reduced forward displacements of cells over time (Fig. 2b, right). Taken together, these dynamic migration data suggest an increased collective cell migration in 2D for ECs lacking BMPR2. This observation may be a consequence of increased cell-to-cell contacts during movement which reduces the overall persistence in migratory directionality and speed of single cells to a level which impedes efficient individual cell forward movement (Fig. 2b, right). Conversely, BMPR2$^{wt}$ cells migrated with lesser interdependency displaying sustained speed and pronounced forward directionality. This becomes particularly clear when seeding both cell types in equal amounts as a dual color mosaic (BMPR2$^{wt}$ in green, BMPR2$^{+/-}$ in magenta) within the same gap-field-of-view (supplementary Movie 2). Here WT cells (green) are preceding BMPR2 deficient cells (magenta) during gap closure suggesting an impaired leader-cell phenotype of BMPR2 deficient cells. These data are further underlined by junctional staining of beta-catenin showing broader junctions in BMPR2$^{+/-}$ cells. This is suggesting stronger cell-cell adhesion participating in hindering coordinated forward movement and decreasing overall migratory capacity (Fig. 2c). Further confocal analysis of filamentous actin (F-actin) stained ECs showed that indeed BMPR2$^{+/-}$ cells display increased cell-to-cell contact sites and lack of free space between cells (Supplementary Fig. 17). Exacerbated collective cell migration of ECs may be a functional consequence of changes in the cell-to-cell adhesion forces[58] facilitating cellular contractility[59,60]. The Rho-ROCK-myosin II pathway acts upstream of actomyosin mediated cell-contractility[61,62]. Investigating the relative localization of active, i.e. phosphorylated myosin light chain (pMLC) to cortical actin bundles at the cells leading edge revealed altered co-localization of pMLC and cortical F-actin in BMPR2$^{+/-}$ cells compared to controls (Fig. 2d). For this, we measured relative intensity profiles from the proximal leading edge towards a more distal side of the cell. In WT ECs, typical cortical F-actin bundles are seen at the cell periphery and the actomyosin rich zone (green) located more inwards within the region of the lamellum (Fig. 2d, left). In contrast, pronounced co-localization of F-actin bundles with pMLC containing actomyosin was found at the very periphery of BMPR2$^{+/-}$ cells, overlapping with the leading edge cell membrane (Fig. 2d, left). This is indicating a change in spatial organization of the contractile actomyosin cytoskeleton in the absence of BMPR2 expression in ECs (Fig. 2d, right). To next gain more information on the adhesive properties of BMPR2$^{+/-}$ cells, we investigated the organization of vinculin-rich FAs, which displayed decreased circularity in case of BMPR2 loss

(Supplementary Fig. 18). However, this phenotype appeared in 2D only when cells were seeded on stiff glass but not on softer polydimethylsiloxane substrates (Supplementary Fig. 18). From this we determined that the role of BMPR2 in modulating EC mechanical behavior during angiogenic processes should be further studied in more physiologically relevant 3D models including a more complex ECM environment.

Albeit not recapitulating essential aspects of angiogenic sprouting such as lumen formation, tube formation is still used widely to analyze the angiogenic potential of cells and stimuli[63–65]. Computational models suggest that in the absence of TC phenotype, ECs still form blood vessel-like structures, however displaying abnormal morphology[66]. We thus speculated that tube capillary network formation may be affected upon BMPR2 deficiency. Accordingly, we performed mosaic tube formation assays by seeding equal amounts of wildtype (green) and BMPR2$^{+/-}$ cells (magenta) together on soft Matrigel. We opted for this mosaic approach to reveal if BMPR2 expression or deficiency promotes certain occupancy of ECs within the tube-network due to different migratory and adhesive properties as suggested by our previous findings. At early time points after seeding, BMPR2$^{+/-}$ ECs (magenta) preceded BMPR2$^{wt}$ ECs (green) in early cell-to-cell contact events (1.5–4 h). These are required for initiation of fusion and first immature capillary tube-network formation. Only at later stages of tube formation (6 h), BMPR2$^{wt}$ ECs also participated in the tube network (Supplementary Fig. 19) particularly when tube-sizes increased and network matured. This suggests that in a softer 3D ECM environment, BMPR2 deficiency affects angiogenesis by increasing cell-to-cell adhesion e.g. during tube network formation.

## BMPR2 regulates spatial organization of pulling forces in nascent sprouts revealed by EC-spheroids combined with traction force microscopy

Since our 2D findings suggested altered actomyosin organization at the leading edge of migrating BMPR2$^{+/-}$ leader cells, we decided to further investigate the mechanical consequences of BMPR2 deficiency, particularly on TC pulling forces during angiogenic sprouting. For this, we applied three-dimensional traction force microscopy (3D TFM) on sprouting ECs originating from spheroids in a fibrin-based hydrogel with a fibrin concentration of 2.5 mg/ml. BMPR2$^{wt}$ EC or BMPR2$^{+/-}$ EC spheroids were embedded in fibrin gel forming higher ordered ECM network[67] (Supplementary Fig. 20). Fibrin hydrogels have superb optical properties for imaging, mimic the in vivo microenvironment of immediate wound healing after blood coagulation and recapitulate major aspects of the transitional ECM in regenerative processes which was shown to provide a beneficial environment for sprouting angiogenesis[68]. The embedding of fluorescent fiducial markers within the gels (Supplementary Fig. 20), subsequently allows the tracking of ECM displacements generated by the pulling forces of nascent sprouts upon experimentally induced cell relaxation using the Rho kinase (ROCK) inhibitor Y-27632. To our knowledge, the combination of using fibrin hydrogels with 3D TFM on EC spheroids has not been performed before (Fig. 2e). ROCK inhibition led to cellular relaxation and sprout collapse with similar kinetics compared to Cytochalasin-D treatment (supplementary Movie 3). ECM displacements from ECM from both BMPR2$^{wt}$ and BMPR2$^{+/-}$ sprouts revealed a typical peak of displacements around sprout tips, where pulling by the TC is expected to occur prominently (Fig. 2e). The observed displacements were in the range of 1–3 μm and in accordance with previous data gained from BMPR2$^{wt}$ ECs sprouting in a related technical setup[41]. The data shown in Fig. 2e suggested a different displacement pattern of BMPR2 deficient sprouts as compared to WT controls. However, we observed a high level of data variability in 3D TFM experiments depending on the morphology of the sprout and the level of sprout matureness at the timepoint of measurement as well as the overall sprout density. We were therefore concerned about the reproducibility of those data and performed >60 experimental replicates per condition to address this assay-dependent variability and to be able to average the data. By analyzing the average 3D TFM displacements along the normalized distance of sprouts of various length, we found that the averaged 3D TFM displacements do not differ between BMPR2$^{wt}$ ECs and BMPR2$^{+/-}$ ECs (Fig. 2f and Supplementary Fig. 21).

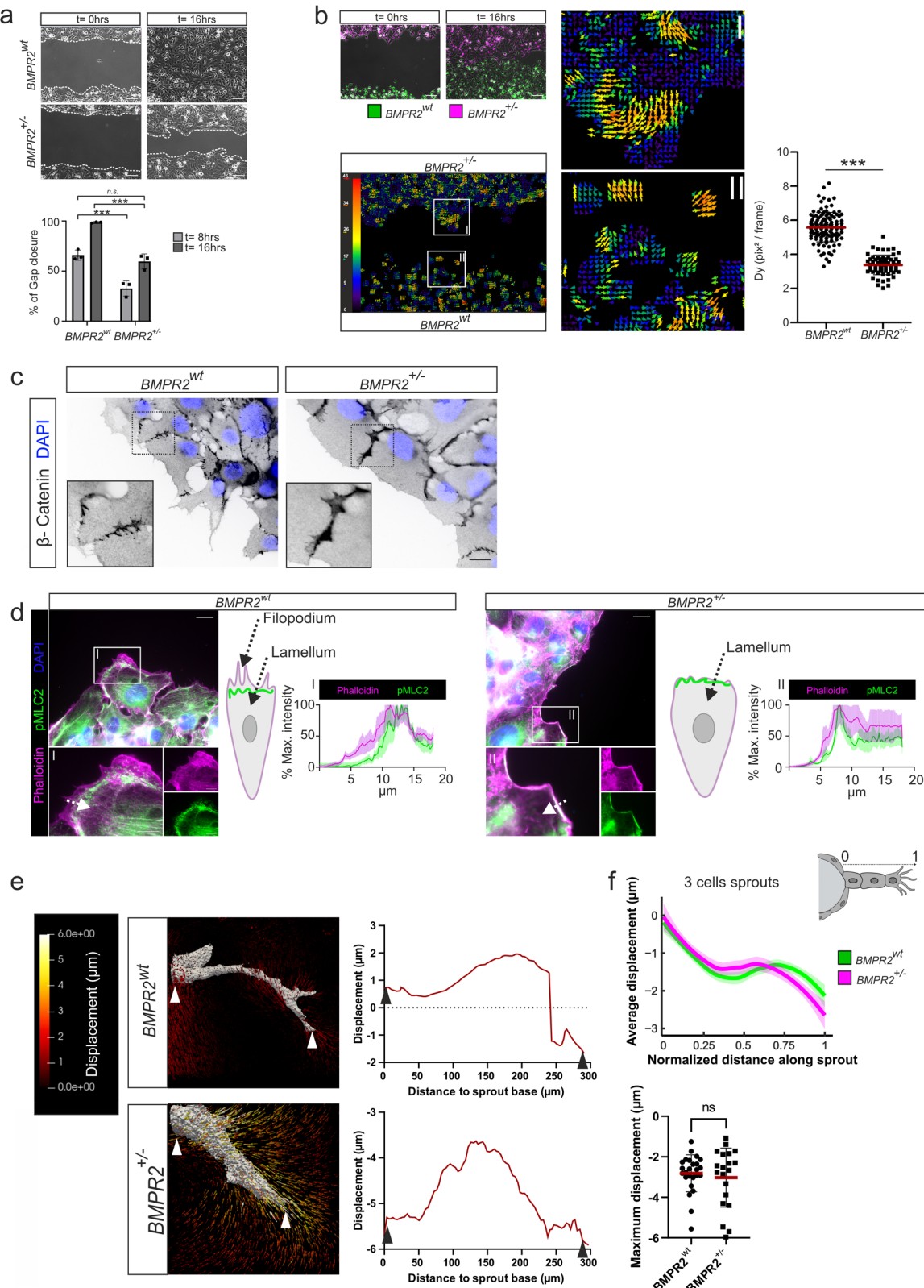

Individual sprouts may be displaying big differences between the two genotypes, however, as averaged dataset, the measured displacements were not revealing a general traction problem of BMPR2-deficient TCs. This result suggests that a loss of BMPR2 does not generally interfere with the ECs ability to pull and interact with the ECM, and that TC pulling is also possible in the absence of BMPR2 expression by the TC.

## BMPR2 is required for efficient sprout outgrowth and determines tip-cell position and shape in EC spheroids

Our previous 2D characterization of BMPR2-deficient ECs suggested impaired phenotypical features of leader cells including lack of filopodia formation, impaired forward movement during cell migration and altered actomyosin organization at the leading edge. While our 3D TFM

**Fig. 2 | BMPR2 is required for polarized EC migration, regulates spatial acto-myosin organization at the EC leading edge and organizes 3D pulling force distribution at the sprout front during angiogenic sprouting in fibrin ECM.**
**a** Phase contrast pictures of gap closure at 16 h (upper) Scale bar: 100 µm: Quantification of gap closure for BMPR2$^{wt}$ or BMPR2$^{+/-}$ ECs at 8 h or 16 h after silicone insert removal. (lower) ***p < 0.001. **b** Images of wound healing assay with BMPR2$^{wt}$ ECs (green) and BMPR2$^{+/-}$ ECs (magenta) at 0 h or 16 h after silicone insert removal. BMPR2$^{wt}$ (bottom) ECs and BMPR2$^{+/-}$ (top) ECs were seeded separately in one compartment of the seeding insert. Scale bar: 100 µm. (middle) Particle image velocimetry analysis of trajectories from ECs used in gap closure assay on the (left). Vectors indicate the main direction and the magnitude of ECs displacements over time. Insets show regions of interest (I & II). (right) Quantification of EC displacement magnitude (D$_y$) measured by trajectory analysis for BMPR2$^{wt}$ ECs and BMPR2$^{+/-}$ ECs and expressed in pixel² per frame. ***p < 0.001. **c** Immunofluorescence staining of BMPR2$^{wt}$ ECs and BMPR2$^{+/-}$ ECs junctions. Beta-Catenin (black), DAPI (blue). Insets show regions of interest. Scale bar: 20 µm. **d** Immunofluorescence staining of BMPR2$^{wt}$ ECs and BMPR2$^{+/-}$ ECs: Phalloidin (magenta), DAPI (blue), phospho-Myosin Light Chain 2 (pMLC2) (green). Insets show regions of interest. Scale bar: 20 µm. Fluorescence intensity profiles of BMPR2$^{wt}$ ECs and BMPR2$^{+/-}$ ECs were measured along the direction of the arrows depicted (cell periphery towards inside of the cell) and averaged for 5 cells. Confidence bands represent standard deviation of the mean. Graphic representation of observed phalloidin and pMLC2 localization at the leading edge for BMPR2$^{wt}$ ECs and BMPR2$^{+/-}$ ECs. The corresponding locations of EC Filopodia and Lamellum at the polarized cells leading edge are indicated (arrows); F-actin (magenta), contractile actomyosin (green). **e** (left) Absolute hydrogel displacement fields for BMPR2$^{wt}$ or BMPR2$^{+/-}$ sprouts. The displacement field magnitude is indicated by color coding (left). (right) Displacement line scans along individual sprouts showing displacements in µm from sprout origin to tip. Negative values correspond to ECM displacements towards the sprout origin. **f** (upper) Average displacements measured along sprouts of similar length (3 cells) for BMPR2$^{wt}$ ECs and BMPR2$^{+/-}$ ECs. The length of each sprout was normalized from 0 (base of the sprout) to 1 (tip of the sprout) to make them comparable. (below) Maximum displacements measured for individual sprouts for BMPR2$^{wt}$ ECs and BMPR2$^{+/-}$ ECs.

experiments did not allow to identify significantly altered TC tractions upon BMPR2 depletion, the 3D-fibrin gel spheroid sprouting assay still proved beneficial to better understand the role of BMPR2 in sprouting angiogenesis, particularly the need for BMPR2 expression by ECs who aim to acquire the TC position in a developing sprout. Therefore, we first quantified and compared the sprouting areas of BMPR2$^{wt}$ and BMPR2$^{+/-}$ spheroids for a period of 48 h. Spheroid sprouting assays showed a reduced outgrowth area for BMPR2$^{+/-}$ cells when compared to the sprouting area of BMPR2$^{wt}$ cells (Fig. 3a and supplementary Movie 4) suggesting a lack of efficient sprout elongation over time. Kinetic analysis of sprouting area progression showed that BMPR2$^{wt}$ cells increase in elongation speed at about 24 h of experimental time when compared to their BMPR2 deficient counterparts (Fig. 3b). These findings were also recapitulated when analyzing the average distance of TCs from their sprout origins (Fig. 3b). Moreover, after 3D TFM data suggested no general traction problem of BMPR2-deficient TCs, we investigated TC morphology more thoroughly. We found that BMPR2 deficient TCs harbor less protrusive structures in 3D in agreement with our 2D data (Fig. 3c). Analysis of TC morphology, as performed by measuring their "solidity"[69] revealed that BMPR2 deficient TCs adopt a conical shape with a "bullet-like" morphological appearance lacking cellular protrusions. During sprouting angiogenesis, it is known that SCs dynamically compete for the TC position over the course of sprouting[70]. Why this "tug-of-war" of SCs and TCs for the relative positions within a sprout is required for efficient sprout outgrowth is not completely understood[69]. Eventually, this is decisive for sprouts to perform efficient sprouting angiogenesis in a complex 3D environment. Thus, we next investigated whether BMPR2-deficient cells that are devoid of filopodia would still acquire TC position when seeded as a 1:1 mosaic with BMPR2$^{wt}$ ECs. Strikingly, after 36 h of sprouting, sprouts from mosaic spheroids (BMPR2$^{wt}$, green; BMPR2$^{+/-}$, magenta) were significantly devoid of BMPR2$^{+/-}$ cells (magenta) in TC position (Fig. 3d, e). Dynamic quantification of sprout composition at 16 h, 36 h and 60 h confirmed that TC position is almost systematically occupied by BMPR2$^{wt}$ ECs (Fig. 3f and supplementary Movie 5).

Together our data show that BMPR2 expression is required for efficient sprouting angiogenesis of EC spheroids in 3D fibrin matrix. While ECs lacking BMPR2 expression would still form some sprouts, the overall number of sprouts, their protrusiveness and their sprouting area are largely reduced indicating lack of efficient sprout outgrowth in the absence of BMPR2 expression. However, during the entire sprouting process the TC position is occupied by BMPR2 expressing TCs which cannot be replaced by a BMPR2 deficient SC when seeded as a mosaic approach. Thus, BMPR2 expression may be required for dynamic SC to TC exchange at the sprout front. We confirmed our experimental setup by inverting the dyes used to label the cells, with BMPR2$^{wt}$ ECs now labeled in magenta and BMPR2$^{+/-}$ cells now labeled in green, to exclude any effects of the dyes on the sprouting behavior and phenotypes observed (Supplementary Fig. 22). For nearly all investigated sprouts, we again found BMPR2 expressing cells occupying the TC position. This is underlining the necessity for BMPR2 expression to acquire TCs characteristics in this experimental setting.

## BMPR2 governs filopodia formation via CDC42 activation through the PI3K signaling pathway

The phenotypical and functional characterization presented in Figs. 1–3 unambiguously demonstrates the indispensability of BMPR2 at the plasma membrane for EC filopodia formation, EC migration and the spatial regulation of actomyosin contractility. These findings also underscore the pivotal role of membrane BMPR2 facilitating ECs to acquire TC position during angiogenic sprouting. However, the underlying molecular mechanism through which BMPR2 acts as a central membrane protein and regulator, upstream of these biochemical and biomechanical processes, remains elusive. Filopodia formation in TCs critically depends on the activity of the small Rho GTPase CDC42, as established by previous studies[71–74]. Consistent with these observations, the EC-specific deletion of CDC42 has been shown to impede filopodia formation and lead to aberrant sprouting angiogenesis[44], a phenotype recapitulated in our BMPR2$^{+/-}$ cell model. This prompted us to hypothesize that BMPR2 might play a role in regulating CDC42 activity in ECs. To investigate this hypothesis, we first examined the spatial correlation between BMPR2 and CDC42 in BMPR2$^{wt}$ ECs overexpressing HA-tagged BMPR2 and show that it locates at the same foci as CDC42 (Fig. 4a, green). This is suggesting spatial proximity and a potential regulatory interplay between these two proteins. Immunostaining revealed that CDC42, like BMPR2, localizes in proximity to filopodial protrusions (Fig. 4a, magenta). To substantiate the notion that BMPR2 is essential for regulating CDC42 activity, we utilized a CDC42 single-chain FRET biosensor, providing a readout of CDC42 activity in living cells[75]. Expression of the CDC42 biosensor in BMPR2$^{wt}$ and BMPR2$^{+/-}$ cells revealed significantly reduced CDC42 sensor activity in BMPR2$^{+/-}$ ECs compared to BMPR2$^{wt}$ controls (Fig. 4b). This biosensor is primarily anchored to the cell plasma membrane via a myristoyl group (Fig. 4b). We further confirmed the involvement of CDC42 in BMPR2-dependent filopodia formation by overexpressing a constitutively active (c.a.) CDC42-GFP fusion protein in BMPR2$^{+/-}$ ECs. The introduction of c.a.-CDC42-GFP successfully induced robust filopodia formation in BMPR2 deficient cells that would otherwise show the typical "blunted" membrane appearance (GFP-only control) (Fig. 4c). The mechanism by which BMPs regulate Rho GTPase signaling including CDC42 activation is not entirely understood. Some data from cell types other than ECs suggest that SMAD signaling may transcriptionally regulate Rho-guanine nucleotide exchange factors (GEFs)[76]. However, our own data (Supplementary Fig. 11) suggested that SMAD signaling may not be involved (Supplementary Figs. 12 and 13). Moreover, previous kinetic studies by others analyzing Rho GTPases activation by BMPs argue for a quick activation within seconds-to-minutes, suggesting no need for transcriptional activity of the BMP pathway[77–80]. To exclude general transcriptional changes in CDC42 expression, we assessed

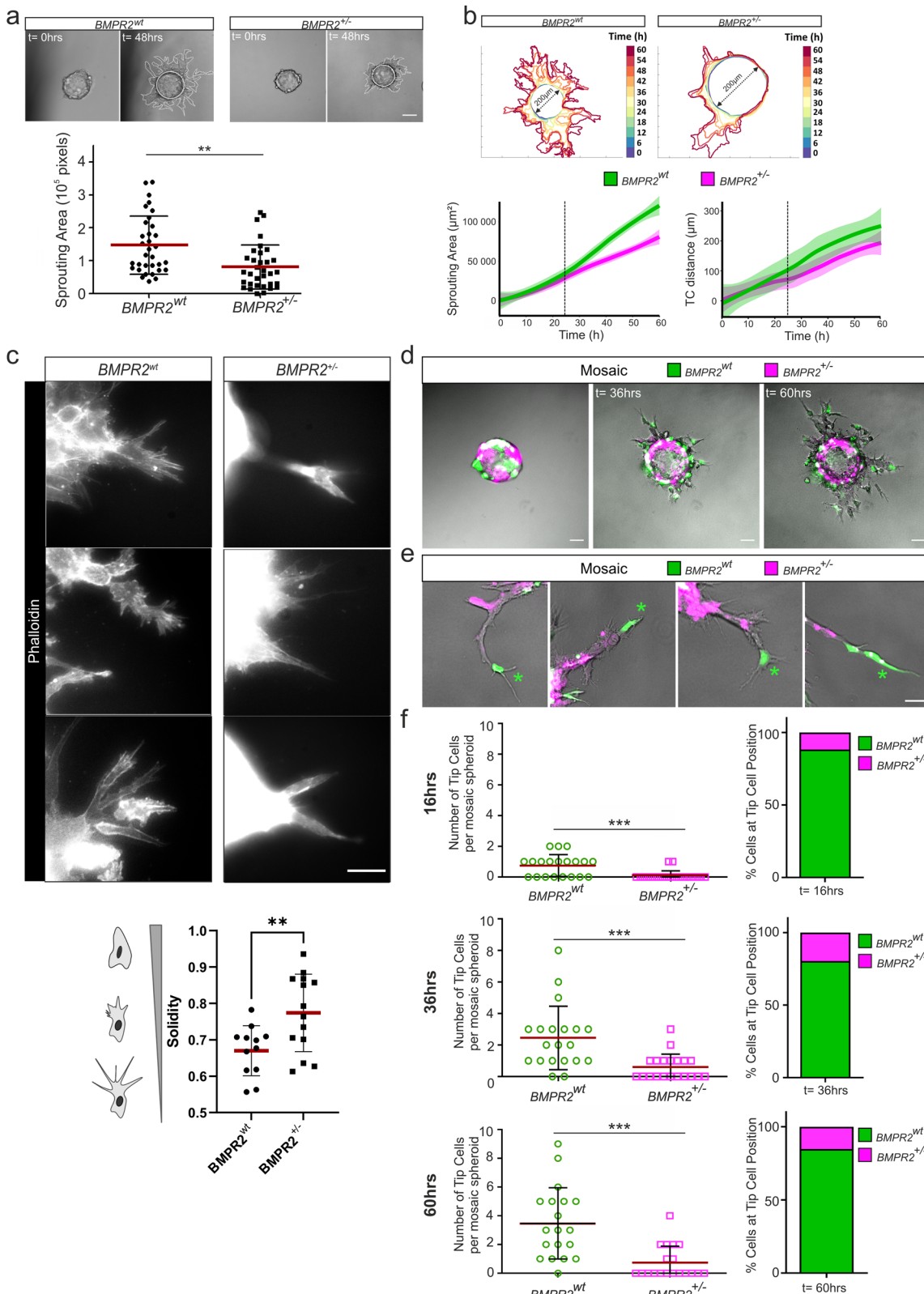

CDC42 protein levels, which showed no change between BMPR2$^{+/-}$ ECs compared to controls (Fig. 4d). Thus, we speculated that BMPR2 dependent regulatory mechanisms independent of gene transcription are acting upstream of CDC42 activity. Along these lines, we previously demonstrated that BMPR2 is a regulator of PI3K activity, leading to BMP-induced phosphatidylinositol-3,4,5-trisphosphate (PIP$_3$) production at the cell's leading edge in the minutes range[21]. Membrane-bound PIP$_3$ serves as a second messenger for Rho-GEF recruitment and subsequent CDC42 activation[81]. Recently, it was found that the strongest GEFs facilitating endothelial CDC42 activity belong to the Pleckstrin Homology (PH) and RhoGEF Domain containing Family G[82] and harbor PH domains required to tether GEFs to PIP$_3$. To confirm alterations in PI3K signaling in BMPR2$^{+/}$

**Fig. 3 | BMPR2 is required for efficient sprouting and tip cell (TC) position in 3D spheroid assays. a** BMPR2[wt] ECs and BMPR2[+/−] ECs were seeded to coat micro-carrier beads, embedded in fibrin gel and covered with EC activation medium for sprouting. Dotted line indicates sprouting area as measured for quantification. Sprouting area after 48 h was measured in pixel with ImageJ and quantified. **\*\*p < 0.01. Scale bar: 100 μm. b** Visualization and quantification of ECs sprouting kinetics over 60h. The sprouting area of EC spheroids from BMPR2[wt] ECs and BMPR2[+/−] ECs at each time point was represented using a heatmap color coding. (time intervals = 6 h). Plots represents average normalized sprouting area over time for each condition (left) and the average migration distance of the tip cell from the surface of the spheroid over time for each condition (right). Standard deviation of the data was represented as confidence bands around the curves. Dotted lines indicate the 24 h time point. **c** Representative immunofluorescence images of 3D sprout ends from BMPR2[wt] ECs and BMPR2[+/−] ECs stained with phalloidin (white) after 64 h of sprouting. Cell morphology and protrusions were assessed by segmenting and measuring the solidity of several tip cells. Scale bar: 50 μm. **d** Mosaic spheroids with BMPR2[wt] ECs (green) and BMPR2[+/−] ECs (magenta) were seeded in fibrin for sprouting and imaged at 12 h, 36 h, and 60 h after embedding. Scale bar: 100 μm. **e** Mosaic magnified sprouts with BMPR2[wt] TCs (green) and BMPR2[+/−] SCs (magenta) upon 60 h of sprouting. Asterisks indicate position of tip cell. Scale bar: 50 μm. **f** The number of BMPR2[wt] TCs (green) and BMPR2[+/−] TCs (magenta) and their relative proportion were quantified for time points 16 h, 36 h and 60 h of sprouting. **\*\*\*p < 0.001.**

ECs, we assessed the phosphorylation of the PH-domain protein AKT, indicative for active PI3K-PIP$_3$ signaling[83]. When cultured in presence of EC activation media, we observed reduced AKT phosphorylation at phosphosites Ser473 and Thr308 in BMPR2[+/−] ECs compared to controls. Furthermore, AKT phosphorylation was further diminished upon treatment with LY294002 but could be rescued using UCL-TRO-1938, a small molecule activator of PI3K p110α subunit, confirming impaired PI3K-PIP$_3$-PH-domain protein signaling under BMPR2 deficiency (Fig. 4e, Supplementary Fig. 23). Consequently, filopodia formation was impaired in BMPR2[wt] ECs treated with LY294002 but could be robustly rescued in BMPR2[+/−] ECs upon PI3K activation using UCL-TRO-1938 (Fig. 4f). This highlights the importance of PI3K signaling in BMPR2-dependent endothelial filopodia formation. Our data therefore indicate that the presence of BMPR2 in the EC plasma membrane positively correlates with CDC42 activation, possibly through a PI3K-PIP$_3$ dependent mechanism.

Amongst all possible mechanisms decisive for BMPR2-dependent CDC42 activation, we were still interested, if some CDC42-relevant regulators and effector proteins were differentially expressed upon BMPR2 deletion. By performing RNA-Seq we found the CDC42 effector protein p21 (RAC1)-Activated Kinase 1 (PAK1), the CDC42-effector protein1 (CDC42EP1) and the CDC42 GEF FGD6, all shown to regulate actin contractility in ECs[84] and to promote filopodia formation[85] to be differentially expressed (Supplementary Fig. 24). Interestingly, CDC42EP1, also known as binder of Rho GTPase 5 (BORG5), was identified to interact with BMPR2 via previous proteomics approaches[86]. In this study novel BMPR2 interacting proteins were revealed. BORG5 binds active i.e. GTP-bound CDC42 to govern actomyosin contractility[87] and promotes angiogenesis by regulating persistent directional EC migration[88]. Consistently, we could successfully confirm the interaction of BMPR2 and BORG5 using proximity ligation assay (PLA) to visualize their interaction in filopodia of HUVECs expressing these tagged proteins (Supplementary Fig. 25). The BMPR2-dependent expression of CDC42-regulating proteins as well as the colocalization of CDC42 and its effector BORG5 with BMPR2 in filopodia strongly supported an interdependency between membrane BMPR2 presence and membrane located CDC42 activation involving additional signaling mediators such as CDC42 activators (PI3K signaling) and CDC42 effectors (BORG5) spatially controlled for filopodia formation. To substantiate the notion that membrane BMPR2 is required for that the activation of CDC42 in endothelial cells, we used a fluorescent dTomato-WASp(CRIB) biosensor capable of binding active CDC42 and visually assess activation of endogenous CDC42 via relocation of the biosensor to the plasma membrane upon activation by PIP$_3$-anchored GEFs[89]. BMPR2[wt] ECs over-expressing the dTomato-WASp(CRIB) biosensor showed enrichment of active CDC42 at sites of filopodia formation at the plasma membrane (Fig. 4g). This local enrichment of active CDC42 at the plasma membrane could not be recapitulated in BMPR2[+/−] ECs, co-expressing GFP only as a control, but was restored and even further exacerbated when BMPR2-GFP was co-expressed additionally to the CDC42 biosensor. Furthermore, sites of enrichment of active CDC42 colocalized with overexpressed BMPR2-GFP at the plasma membrane, strongly suggesting that BMPR2 spatially regulates CDC42 activation at the plasma membrane of endothelial cells.

In summary our data suggest a mechanistic model where BMPR2 emerges as a pivotal regulator of filopodia formation, thereby playing an indispensable role in tip cell functionality and positioning during angiogenic sprouting. By modulating both the upstream and downstream regulators of CDC42, BMPR2 acts as a crucial mediator of actin and actomyosin related processes at the plasma membrane. Importantly, our study underscores the necessity of membrane BMPR2 for maintaining proper endothelial cell operations, by signaling which is distinct from its traditional role of BMPR1 transactivation. This signaling includes regulation of filopodia formation involving the regulation of CDC42 activity, the organization of the contractile actomyosin at the leading edge, focal adhesion formation and TC morphology in 3D fibrin gels (see Fig. 5; graphical summary).

## Discussion

Prior research has highlighted the pivotal role of BMPs in endothelial cell (EC) activation and sprouting angiogenesis, yet the distinct contribution of the type II receptor BMPR2 and the involvement of non-canonical signaling pathways remain relatively unexplored. Inspired by the work of Wakayama et al. 2015[43] on BMP-induced CDC42 signaling and its requirement for sprouting angiogenesis at the zebrafish caudal vein plexus, we highlight here the important role of BMPR2 in sprouting angiogenesis by human ECs. Our study refines the existing understanding of BMPR2 specific function in this process, shedding light on previously underestimated cellular aspects such as cellular biomechanics and cytoskeletal dynamics influenced by BMPR2 signaling.

To prove that BMPR2 is involved in EC filopodia formation, we used a genetically modified BMPR2-Knock-Out EC model and transient knock-down of BMPR2 in primary ECs. We also performed rescue approaches by reintroducing BMPR2 via overexpression (Fig. 1e, f; Supplementary Figs. 8–15). Of note, overexpression or depletion of a BMP receptor such as BMPR2 may change the intracellular equilibrium of BMP-signaling and the amount of related signaling molecules with unknown consequences. To overcome this hurdle, we have employed multiple compensation strategies (Fig. 1g, 4a; Supplementary Fig. 12–15) and validated our findings on BMPR2 depletion and overexpression in two independent EC models. (Fig. 1f, g). One of our key findings is the indispensable role of BMPR2 in filopodia formation. Previous studies have suggested that BMPR2-deficient microvascular ECs suffer from defective F-actin stress fibers[90], a hallmark of disturbed cell adhesion and signaling processes related to polymerization and bundling of actin. Intriguingly, while our microscopy localization studies did not reveal specific enrichment of BMPR2 in filopodia, single particle tracking microscopy (SPTM) revealed BMPR2 presence in confined foci at the cell periphery, indicative of ligand-bound BMPR2-containing signaling complexes (BISCs; Fig. 1h, Supplementary Fig. 16). This is in support of our previous model established in other cell types, where BMP-induced confinement of BMPR2 contributes to non-canonical MAPK signaling[54,91]. These confined BMPR2 molecules may form part of BISC signaling complexes in proximity to effectors like CDC42 and interact with BORG5, locally limiting BISC activity towards the cytoskeleton at specific membrane sites with microenvironments permissive for filopodia formation[92]. Nevertheless, whether and which BMP ligands are involved in the BMPR2-driven filopodia formation described in this study remains unclear and would need further investigation. Furthermore, we did not find significant differences between the length of the few remaining filopodia of BMPR2 KD ECs and those of control ECs (Supplementary Fig. 7). Finally, we could show

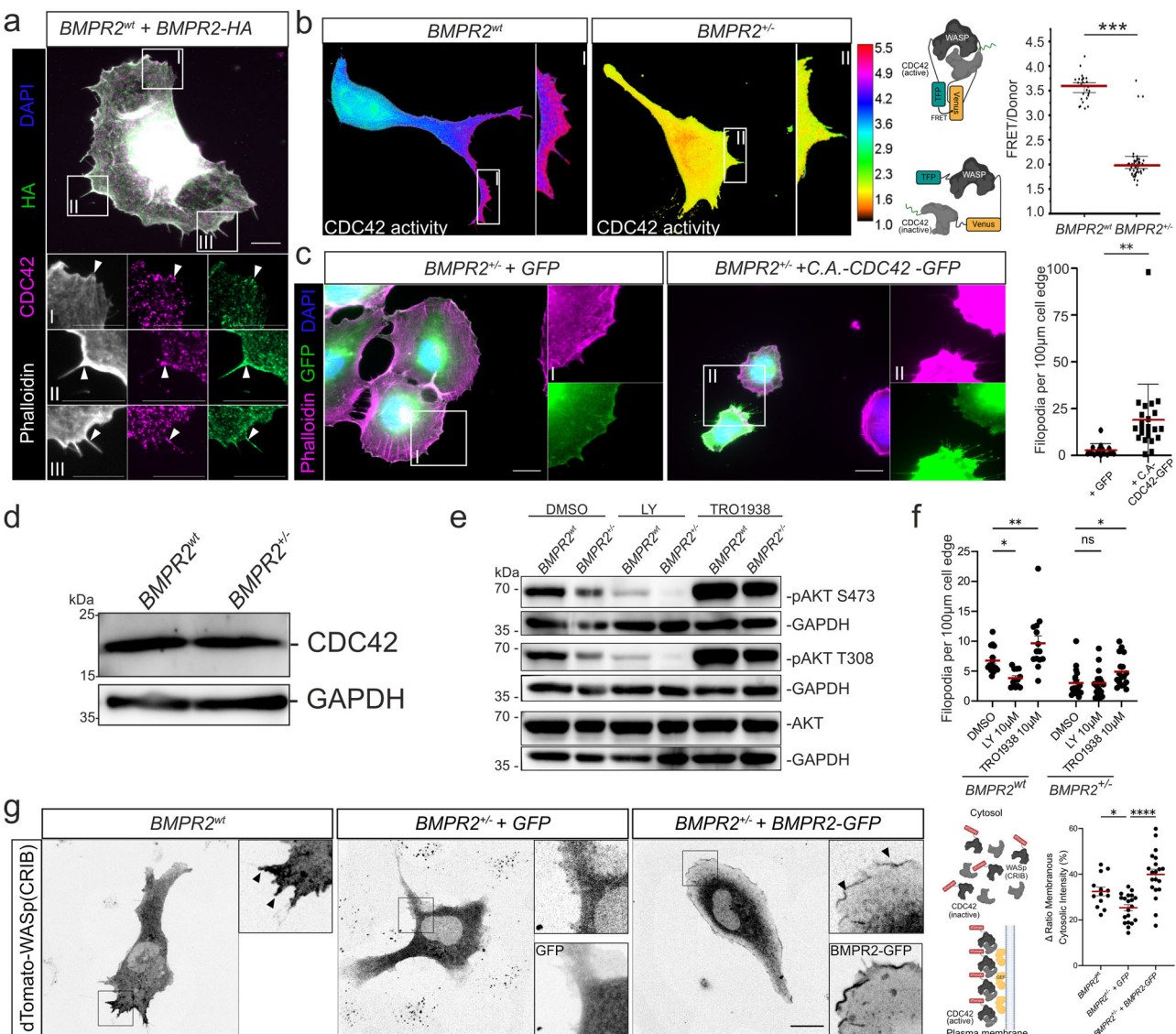

**Fig. 4 | BMPR2 promotes CDC42 activity at the plasma membrane of ECs via the PI3K-CDC42 signaling axis. a** Immunofluorescence staining of BMPR2$^{wt}$ ECs overexpressing HA-tagged BMPR2 (BMPR2-HA). Phalloidin (white), DAPI (blue), BMPR2-HA (green), CDC42 (magenta). Insets show regions of interest. Arrows indicate sites of colocalization between BMPR2-HA and CDC42 at cell cortex or in filopodia compartment base and shaft. Scale bar: 10 µm. **b** Heat-map showing FRET-measured CDC42 activity across whole areas of BMPR2$^{wt}$ ECs (left) and BMPR2$^{+/-}$ ECs (right). Insets show regions of interest. Quantification of FRET signal from depicted TFP-WASP-Venus-CDC42 construct in BMPR2$^{wt}$ ECs and BMPR2$^{+/-}$ ECs (right). ***$p < 0.001$. **c** Immunofluorescence staining of BMPR2$^{+/-}$ ECs over-expressing GFP or a constitutively active CDC42 in fusion with GFP (C.A-CDC42-GFP) and quantification of the number of filopodia per 100 µm of cell edge (right). Phalloidin (magenta), DAPI (blue), GFP (green). Insets show regions of interest. **$p < 0.01$. Scale bar: 20 µm. **d** Immunoblot against CDC42 and GAPDH from BMPR2wt ECs and BMPR2+/- ECs cultured in EC activation medium containing 20% Serum and pro-angiogenic growth factors. **e** Immunoblot against pAKT-Ser473

(S473), pAKT-Thr308 (T308), total AKT (tAKT) and GAPDH from BMPR2$^{wt}$ ECs or BMPR2$^{+/-}$ ECs treated with either DMSO, 10 µM LY294002 or 10 µM UCL-TRO-1938 for 60 min in EC activation medium containing 20 % Serum and pro-angiogenic growth factors. **f** Quantifications of the number of filopodia per 100 µm cell edge of BMPR2$^{wt}$ ECs or BMPR2$^{+/-}$ ECs treated with either DMSO, 10 µM LY294002 or 10 µM UCL-TRO-1938 for 60 min in EC activation medium. *$p < 0.05$, ***$p < 0.001$. **g** Spinning disk microscopy images of BMPR2$^{wt}$ ECs or BMPR2$^{+/-}$ ECs expressing dTomato-WASp(CRIB) active CDC42 biosensor, and co-expressing either GFP or BMPR2-GFP for BMPR2$^{+/-}$ ECs. Scheme depicts the mechanism of action of the CDC42 activity biosensor: Upon activation by PIP$_3$-anchored GEFs, CDC42 is enriched at the plasma membrane and can be bound by dTomato-WASp(CRIB) biosensor, promoting relocation and local enrichment of the fluorescent biosensor. Relocation of the dTomato–WASp(CRIB) biosensor was measured by quantifying the ratio of the sensor membranous intensity over its cytosolic intensity. Arrowheads indicate sites of enrichment of the biosensor at the plasma membrane. Scale bar: 20 µm. *$p < 0.05$, ****$p < 0.0001$.

that BMPR2 needs to have an active kinase domain to successfully promote filopodia formation in ECs (Supplementary Fig. 15). Taken together, this suggests that BMPR2 is an important upstream regulator initiating the formation of endothelial filopodia but not regulating their elongation, while at the same time BMPR2 may not being the only protein covering this function in ECs.

To investigate potential alterations in cell mechanics, we used the phosphorylated form of myosin light chain as an indicator of the spatial

organization and contractility of actomyosin at the cells' leading edge[93]. Our data revealed that BMPR2 deficiency led to altered localization of acto-myosin at the cell cortex when compared to WT cells, resulting in exacerbated 2D collective sheet migration, changes in focal adhesion (FA) and adherens junctions' assembly, and a different temporal behavior of BMPR2 deficient cells during tube formation and sprouting. Previous studies have shown that a contractile actomyosin located at the cell cortex as the one reported in our BMPR2 deficient ECs can cause an increase in plasma

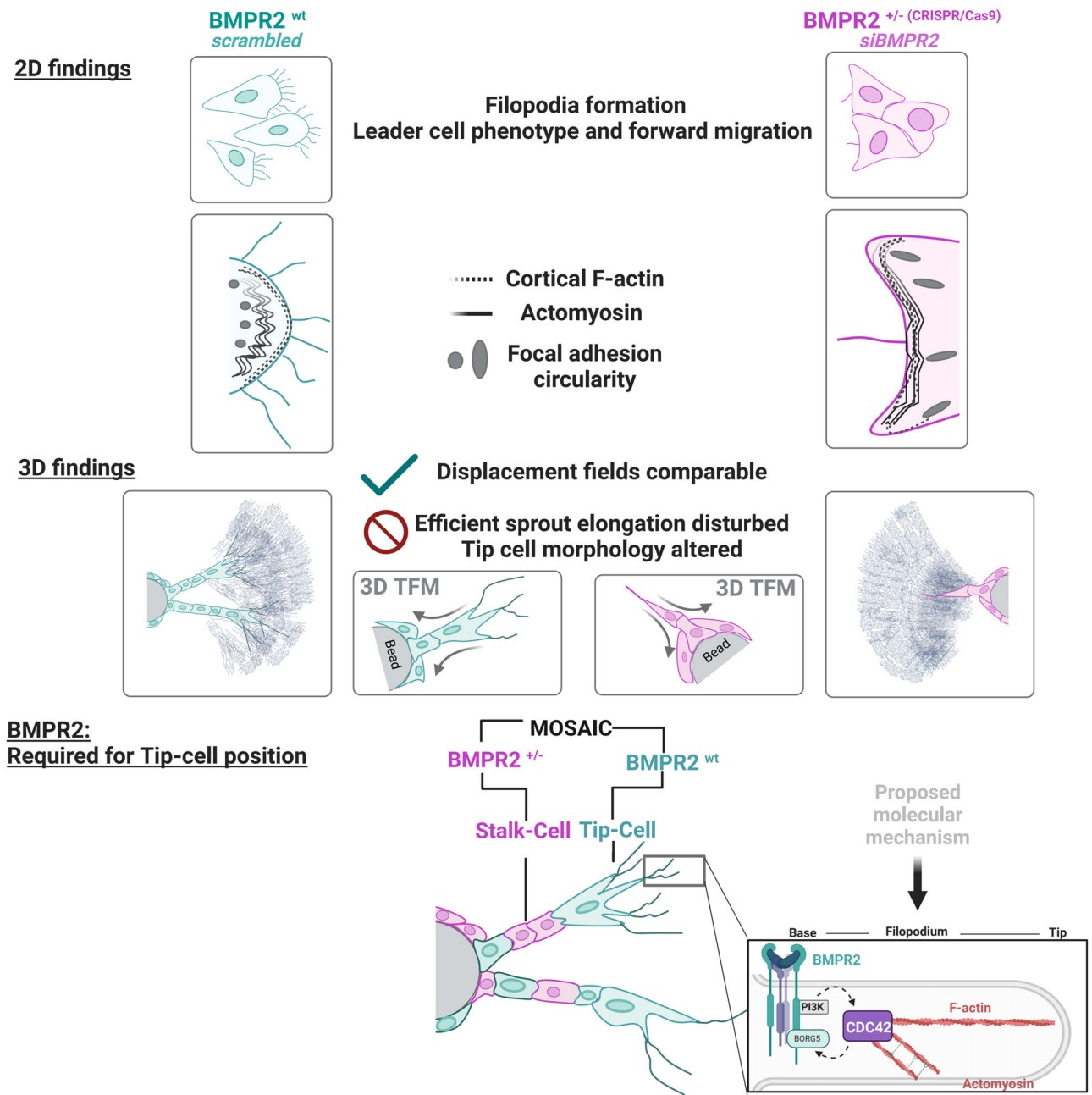

**Fig. 5 | Graphical summary.** Endothelial cells form filopodia to sense their environments and facilitate chemotaxis-induced migration. In 2D migration assays, BMPR2-deficient cells fail to form filopodia and display an exacerbated collective migration behavior upon lack of BMPR2 expression which impedes with their efficient forward movement. We also found an increased co-localization of actomyosin together with cortical actin at the leading edge of BMPR2 deficient cells. Also, focal adhesion formation is altered. In 3D sprouting assays, BMPR2 deficiency stalls sprouting and abrogates ECs from acquiring the tip cell (TC) position. Wildtype and BMPR2 deficient sprouts perform pulling which is dominated by the tip-cell (TC), while we find no general change in overall pulling forces between BMPR2[wt] and BMPR2[+/-] sprouts. In spheroid sprouting assays as mosaic, loss of BMPR2 expression by the TC identifies *BMPR2* as a gene required for acquiring TC position in competition with stalk cells (SCs). In our proposed molecular model, we identified BMPR2 to regulates CDC42 activity. We propose that CDC42 dependent actin polymerization is facilitated in proximity to filopodia by ref. 1 inducing non- canonical PI3K signaling and[2] via yet to be confirmed BMPR2 interacting CDC42 effector protein BORG5, known to promote local actomyosin contractility in ECs.

membrane tension[94]. In turn, the increased in-plane membrane tension at the cell leading edge limits the formation of filopodia and promotes the retraction of existing ones[95]. Thus, the shift towards a cortical localization of actomyosin in BMPR2 deficient ECs suggests an increased membrane tension contributing to the lack of filopodia.

Alterations in cell-to-substrate adhesion are likely to impact cellular forces and vice versa. We thus examined force-induced matrix displacements exerted by the tip cells (TCs) on the extracellular matrix (ECM) in the absence of filopodia and filopodia-associated FAs. Previous studies have established a clear picture on how pulling forces are typically distributed along a normal angiogenic sprout[41]. We were able to recapitulate those findings and show that both BMPR2[wt] and BMPR2[+/-] sprouts display a typical peak of displacements around sprout tips. However, to our surprise and after performing extensive 3D TFM analysis, we ultimately could not detect a significant difference in the overall pulling pattern of BMPR2-deficient cells compared to control ECs. BMPR2 deficiency could possibly become temporarily compensated by other type II receptors depending on their dynamic expression levels during sprouting angiogenesis. However,

while validating previous 3D TFM findings, those sprouts were also analyzed to assess the relative position of ECs as TC or SC within a sprout using a mosaic approach. This revealed a clear necessity for BMPR2 expression by ECs to acquire the TC position. Similar functions to the one that we report here for BMPR2 were described for vascular endothelial growth factor receptor 2 (VEGFR2)[40,96] in concert with VEGFR- co-receptor neuropilin-1 (Nrp1)[37,97]. Both were shown to participate in FA assembly, integrin signaling and activation of CDC42 and are appreciated as robust TC markers[40,71,98]. An angiogenic zone that is BMP dependent but VEGF independent, such as the zebrafish CVP, is not known in humans. It is therefore possible that mechanisms involving VEGF-VEGFR2 signaling are able to partially compensate for BMPR2 deficiency in EC filopodia formation in higher vertebrates.

Finally, our study demonstrated that BMPR2 deficiency impairs PI3K-$PIP_3$ signaling, a vital pathway for recruiting and activating proteins involved in filopodia formation at the cell membrane. We propose that BMPR2 indirectly activates CDC42, with BMPR2-PI3K signaling promoting the generation of $PIP_3$ acting as a membrane-bound second messenger upstream of CDC42 activation. Furthermore, BMPR2 depletion resulted in the dysregulation of CDC42-related regulators and effectors. Importantly, our research confirmed the co-localization of BMPR2 with CDC42 and the CDC42 effector protein BORG5 at sites of filopodia formation. Moreover, CDC42 FRET biosensors and endogenous activity biosensors revealed that BMPR2 depletion results in a significantly decreased CDC42 activity. Together these findings underscore the crucial role of BMPR2 in promoting CDC42 activity upstream of endothelial filopodia formation essential for TC positioning and sprouting angiogenesis.

In conclusion, we have demonstrated that BMPR2 serves as a central integrator of endothelial tip-cell filopodia formation and cell mechanics, proposing a molecular mechanism involving the regulation of CDC42 upstream of these cellular functions. In the future, technical challenges that permit us to combine and correlate different sensors with the here established 3D in vitro spheroid culture will have to be overcome. This will allow to gain more insights into the spatial organization, localization, kinetics, activity, and dependencies of BMPR2 and associated proteins during the process of sprouting angiogenesis.

## Material and methods
### Cell culture
EAhy926 BMPR2[wt] and BMPR2[+/−] cells were generated and validated as described previously[48]. HUVECs were a kind gift from M. Lorenz, V. Stangl and A. Pries (Berlin, Germany) and for expansion cultured on gelatin-coated tissue culture ware (Greiner Bio-One). Ethics approval and consent to participate: Isolation of HUVEC conformed to local university guidelines and with the principles outlined in the Declaration of Helsinki (Approvement by the Charité University Hospital Ethics Committee, EA2/017/13)". EAhy926 cells and HUVECs were maintained and expanded in M199 medium (Sigma Aldrich) supplemented with 20% FCS (Biochrom, Germany), 2 mM L-glutamine (PAN-Biotech, Germany), 100 units/ml penicillin, 10 µg/ml streptomycin (PAA Laboratories), 25 µg/ml Heparin (Sigma Aldrich), and 50 µg/ml EC growth supplement/ ECGS (Corning, NY), referred to as "EC activation medium" recapitulating plasma mix of TGFbeta growth factors and other pro-angiogenic factors[99–101]. All cells were cultured at 37 °C and 5% $CO_2$ atmosphere. For single cell analysis cells were imaged at maximum density of 50%. For maintenance and expansion, typically 500,000 cells were seeded in T75 TC plastic flasks (Greiner Bio-One International) and passaged using Trypsin (PAN-Biotech, Germany) (unless stated otherwise) in 3–4 d intervals, with cells reaching confluence after day 2–3 upon seeding. For sprouting angiogenesis assays (unless stated otherwise), we seeded 15,000 cells per well in a 12-well plate (or coverslip in same format) with cells reaching about 60% sub-confluence 24 h later. Starvation of cells was carried out after rinsing cells in phosphate buffered saline (PBS) (PAN-Biotech GmbH, Germany) and exposure to starvation medium (M199 media) containing 100 units/ml penicillin, 10 µg/ml streptomycin, and 25 µg/ml Heparin for 6 h. Automated cell count was conducted using CASY Model-TT cell-analyzer (Roche, Germany) and included monitoring of cell viability and proliferation capacity from passage 4 to passage 30 for cell lines and passage 1–3 for primary HUVEC. Cos7 cells (ATCC) were maintained and expanded in DMEM (Biochrom, Germany) containing 1.0 g/l D-glucose, 2 mM L-glutamine, 100 units/ml penicillin, 10 µg/ml streptomycin, and 10% FCS. Table 1

Ethics approval and consent to participate: The HUVEC primary cells were isolation from donors conformed to local university guidelines and with the principles outlined in the Declaration of Helsinki (Approvement by the Charité University Hospital Ethics Committee, EA2/017/13) and not sex typed. The detailed isolation procedure can be found in Mendez P…. Knaus P. et al. BMC Biol 2022 (PMID: 36171573).

### Drug treatments of cells and spheroids
Pharmacological inhibitors LDN-193189 (5 µM), LY294002 (10 µM) and small molecule activator UCL-TRO-1938 (10 µM) were added in EC activation medium containing 20% Serum and pro-angiogenic growth factors for 60 min prior to lysis. Y-27632 was added to cells imaged at the microscope and cell relaxation was monitored online. Equal volumes of dimethylsulfoxide were used to treat control cells.

### Starvation and BMP6 treatment of ECs
Production of recombinant human BMP6 was previously described[102]. Unless indicated otherwise, ECs were starved for 4 h before stimulation in M199 medium (with Earle's salts and NaHCO3) that was supplemented with 2 mM L-glutamine, 100 U/ml penicillin, and 0.1 mg/ml streptomycin, herein referred to as endothelial starvation medium. For all assays, following starvation time, cells were pretreated with small molecules inhibitors or activators as indicated for 60mn prior to concomitant stimulation with 10 nM BMP6. For immunoblotting, cells were stimulated with BMP6 for 60mn before lysis. For immunofluorescence, cells were stimulated with BMP6 upon insert removal and fixed after 3 h.

### siRNA and expression plasmid transfection
Transfection of Cos7 cells was carried out using the polyethylenimine (PEI) method as previously described[103]. Transfection of Eahy926 cells and HUVECs was performed using Lipofectamine2000 (ThermoFisher Scientific) according to manufacturer instructions. Typically, 1–5 µg of plasmid were expressed in cells covering an area of 3 cm². Cells were left after transfection for protein expression for 12–14 h. For transient knock-down of BMPR2, Accell siRNA SMARTPool targeting BMPR2 (Dharmacon, E-005309-00-0010) were used at a concentration of 1 µM diluted in Accell Delivery Media (Dharmacon, B-005000) as indicated by the manufacturer. A table of expression plasmids and siRNAs used in this study is shown below.

### Generation of Bmpr2 depleted zebrafish embryos by morpholino targeting and CRISPR-Cas9 technology
Handling of zebrafish was done according to FELASA guidelines (Aleström et al., 2020), in compliance with German and Brandenburg state law, carefully monitored by the local authority for animal protection (LAVG, Brandenburg, Germany, Animal protocol #2347-43-2021). Zebrafish of Tg(kdrl:EGFP)[104] were maintained under standard conditions as previously described[105]. The morpholino targeting exon2 of bmpr2b (5′-AGCTGCCGACACACAA AATGAGAAA-3′ from GENE TOOLS, at 0.3 mM) was injected into the yolk at the one-cell stage in 1 nl total volume. Similarly, for bmpr2b-crispant generation, 1 nl of gRNA-Cas9 mix following the protocol version 3 from Kroll et al.[47], was injected into the cell at one-cell stage. gRNA´s were targeting exon 1 (5′- TGAGAGTGCGGATCCTAAAC – 3′) and exon 3 (5′-GCGT TGTGTGGTAACCACGCAA -3′). Embryos were fixed at the desired stages in 4% paraformaldehyde overnight at 4 °C and washed and kept in PBST. The developmental stage of the embryos used is indicated for each experiment in the results and figure legends.

## Table 1 | Key resource table referring to individual key resources used in this study

**Key resource table**

| Reagent type or resource | Designation | Source or reference | Identifiers | Additional information |
|---|---|---|---|---|
| *Primary cells and Cell lines* | | | | |
| Human EAhy926 | BMPR2^wt^; WT ECs; WT | 48 | | See supporting information[48] Figure S1 and corresponding Materials and Methods for further information |
| Human EAhy926 BMPR2 KO1 | BMPR2^-/-^; BMPR2 deficient ECs; ECs with BMPR2 deficiency | 48 | | Haploinsufficiency model for BMPR2, created by CRISPR/Cas9 in Hiepen et al. Plos Biol. 2019; see supporting information[48] Figure S1 and corresponding Materials and Methods for further information |
| Human umbilical vein endothelial cells (female) | HUVEC | 113 | | Single donor cells were sex-specifically genotyped prior to application in this work |
| African green monkey kidney fibroblasts (SV40 T antigen) | COS7 | ATCC | CRL-1651 | |
| *Animal models* | | | | |
| Tg(kdrl:EGFP) ^s843^ | Tg(kdrl:EGFP) | 104 | ZFIN:ZDB-ALT-050916-14 | |
| *Transfected constructs* | | | | |
| pcDNA3.1 BMPR2-HA- LF | BMPR2-HA | 48 | | Long form of BMPR2 amino- terminally tagged with HA |
| pcDNA3.1 BMPR2-HA- SF | BMPR2-HA-SF | 48 | | Short form of BMPR2 amino- terminally tagged with HA |
| pcDNA3.1 BMPR2-MYC-K230R | BMPR2-K230R-Myc | | | |
| pcDNA3.1 ALK1-meGFP | ALK1-GFP | | | |
| pcDNA3.1 ALK2-meGFP | ALK2-GFP | | | |
| pcDNA3.1 ALK3-meGFP | ALK3-GFP | | | |
| pcDNA3.1 Endoglin-meGFP | Endoglin -GFP | | | |
| BMPR2 c-term myc | BMPR2-myc | 48 | | The coding sequence of human BMPR2 was C-terminally myc--tagged and subcloned into pcDNA3.1 from HA-BMPR2-LF plasmid |
| BMPR2- n-term EGFP | BMPR2-GFP | this paper | | Monomeric enhanced GFP cloned after signal peptide. Gly5 linker between megfp and mature LF BMPRII sequence. |
| pRK5 meGFP | GFP | | Addgene #18696 | |
| CDC42-Q61L-eGFP | C.A. CDC42-GFP | 114 | | Activating mutation at position 61 |
| pTriEx4-Cdc42-2G | CDC42- biosensor; CDC42 activity sensor; CDC42 2 G activity sensor | 115 | Addgene #68814 | Second generation CDC42 activity sensor based on original sensor design from Klaus Hahn lab but with modified domain orientation and permutated fluorescent proteins for better dynamic range |
| Lifeact-TagGFP2 | LifeAct-GFP | Ibidi USA; Fisher scientific | #50-114-9044 | |
| CMVdel-dimericTomato-WASp(CRIB) | Active endogenous CDC42 biosensor | 89 | Addgene #191450 | |
| *Oligonucleotides used in this study* | | | | |
| Morpholinos used | | | | |
| *bmpr2b* E2 MO | | GENE Tools | | 5'AGCTGCCGACACACAAAATGAGAAA-3' |
| gRNA exon 1 | | Integrated DNA Technologies | | 5'- TGAGAGTGCGGATCCTAAAC – 3' |
| gRNA exon 3 | | | | 5'-GCGTTGTGTGGTAACCACGCAA -3' |

**Table 1 (continued) | Key resource table referring to individual key resources used in this study**

| Key resource table | | | | |
|---|---|---|---|---|
| Reagent type or resource | Designation | Source or reference | Identifiers | Additional information |
| siRNA Oligos | | | | |
| Accell siRNA SMARTPOOL targeting BMPR2 | siBMPR2 | Dharmacon | E-005309-00-0010 | |
| Accell FITC-labeled non-targeting siRNA | si-FITC | Dharmacon | D-001950-01-20 | |
| Accell ONTARGETplus non-targeting siRNA | siscr | Dharmacon | D-001920-01-50 | |
| Accell siRNA delivery media | | Dharmacon | B-005000 | |
| ZF-Bmpr2b_E2_rev1 | 5'-CACGATAGCCAAAGAACAGCG-3' | | | |
| ZF-b-actin_fwd | 5'-TGTTTTCCCCTCCATTGTTGG-3' | | | |
| ZF-b-actin_rev | 5'-TTCTCCTTGATGTCACGGAC-3' | | | |
| ZF-bmpr2b-Exon1_fwd | 5'-ATGAGCTTGAACTTCGGTCGAT-3' | | | |
| ZF-bmpr2b-Exon1_rev | 5'-GACACCCTTTAAGACATTGCAGG-3' | | | |
| ZF-bmpr2b-Exon3_fwd | 5'-GTATCTCAGTGTTTCACTCCACC-3' | | | |
| ZF-bmpr2b-Exon3_rev | 5'-ATGACACATCACAACATGCTGATACA-3' | | | |
| *Antibodies used* | | | | |
| anti-phospho SMAD1/5 (Ser463/465) (41D10) | | Cell Signaling Technology | #9516 | Dilution WB: 1:1000 |
| Anti-Smad1 XP | | Cell Signaling Technology | #6944 | Dilution WB: 1:1000 |
| anti-phospho-AKT (Ser473) | | Cell Signaling Technology | #9271 | Dilution WB: 1:1000 |
| anti-phospho-AKT (Thr308) | | Cell Signaling Technology | #9275 | Dilution WB: 1:1000 |
| anti-AKT | | Cell Signaling Technology | #9272 | Dilution WB: 1:1000 |
| anti-phospho MLC2 (Thr18/Ser19) | | Cell Signaling Technology | #3674 | Dilution WB: 1:1000 |
| anti-GAPDH | | Cell Signaling Technology | #14C10 | Dilution WB: 1:1000 |
| anti-Myc | | Cell Signaling Technology | #9B11 | Dilution IF: 1:10.000 Dilution PLA: 1:5000 |
| anti-GFP | | Cell Signaling Technology | #2956 | Dilution PLA: 1:500; IF: 1:750 |
| anti-HA (HA-7) | | Sigma-Aldrich | | Dilution SPTM: 1 µg/ml; IF: 1:1000 |
| anti-vinculin | | Sigma-Aldrich | #V9131 | Dilution IF: 1:400 |
| anti-Beta-Catenin | | BD Bioscience | #610153 | Dilution IF: 1:200 |
| anti-BMPR2 | | BD Bioscience | #612292 | |

**Table 1 (continued) | Key resource table referring to individual key resources used in this study**

**Key resource table**

| Reagent type or resource | Designation | Source or reference | Identifiers | Additional information |
|---|---|---|---|---|
| anti-CDC42 | | Abcam | #155940 | Dilution WB: 1:1000 |
| anti-CD34 | | Abcam | #81289 | Dilution IF: 1:750 |
| Anti-Mouse Alexa Fluor 488 (goat) | | Thermofisher Scientifics | A10684 | |
| Anti-Mouse Alexa Fluor 594 (goat) | | Thermofisher Scientifics | A11020 | |
| Anti-Rabbit Alexa Fluor 488 (chicken) | | Thermofisher Scientifics | A21441 | |
| Anti-Rabbit Alexa Fluor 594 (goat) | | Thermofisher Scientifics | A11072 | |
| Goat anti rabbit 2ndary HRP-linked antibody | | Dianova | 111-035-144 | Pan BMP-Smad inhibition |
| Goat anti mouse 2ndary HRP-linked antibody | | Dianova | 115-035-068 | Pan PI3K inhibition |
| ***Critical commercial assays*** | | | | |
| Proximity Ligation Assay (PLA) DUOLINK kit | | Sigma-Aldrich | #DUO92101 | Rho Kinase inhibition |
| ***Dyes (conjugated)*** | | | | |
| DAPI | | Sigma-Aldrich | D954 | |
| Phalloidin CruzFluor 594 conjugate | | Santa Cruz Bio | sc-363795 | |
| Phalloidin CruzFluor 647 Conjugate | | SantaCruz | Sc-363797 | |
| Vybrant™ DiI Cell Labeling Solution | | Thermo- Fisher Scientific | V22885 | Vybrant™ DiI Cell Labeling Solution |
| Vybrant™ DiO Cell Labeling Solution | | Thermo- Fisher Scientific | V22886 | Vybrant™ DiO Cell Labeling Solution |
| Celltracker Red CMTPX Dye | | Thermo- Fisher Scientific | C34552 | |
| Celltracker Green CMFDA Dye | | Thermo- Fisher Scientific | C2925 | |
| ***Small Molecule Inhibitors*** | | | | |
| LDN-193189 | LDN | Tocris Bioscience | # 6053 | |
| LY294002 | LY | Cell Signaling | # 9901 | |
| UCL-TRO-1938 | TRO1938 | MCE | HY-154848 | |
| Y-27632 | Rocki; Rock inhibitor | Stemcell Technologies | # 72304 | |
| CytochalasinD | CytoD | Thermofisher Scientifics | PHZ1063 | |
| ***Critical chemicals used in this study*** | | | | |
| Dimethyl sulfoxide (DMSO) | | Sigma-Aldrich | D2650 | |
| Protease Inhibitor Cocktail | | Sigma-Aldrich | P6148 | |

**Table 1 (continued) | Key resource table referring to individual key resources used in this study**

| Reagent type or resource | Designation | Source or reference | Identifiers | Additional information |
|---|---|---|---|---|
| **Key resource table** | | | | |
| Paraformaldehyde 4% (PFA) | | VWR | VWRK4186 | |
| ***Purified proteins and additives*** | | | | |
| Thrombin | | Sigma-Aldrich | T4648 1KU | |
| Fibrinogen type 1 | | Sigma-Aldrich | F8630 | |
| Endothelial cell growth supplement (ECGS) | ECGS | Corning, NY | #356006 | |
| Heparin | | Merck | 375095-100KU | |
| Fibronectin from bovine plasma | | Sigma-Aldrich | F1141 | |
| Normal goat serum | | Sigma-Aldrich | NS02L | |
| Aprotinin | | Sigma-Aldrich | A1153 | |
| Recombinant hBMP6 | BMP6 | 102 | | Human mature BMP-6 produced in Chinese hamster ovary cells |
| ***Other research tools*** | | | | |
| Fluospheres Dark Red | | Thermofisher Scientifics | F8807 | |
| Quantum dot labeled F(ab')2-Goat anti-Mouse IgG (H + L) Secondary Antibody, Qdot 655 nm | Qdot antibody | Thermo- Fisher Scientific | #Q-11021MP | Further information on QDot labeling and SPTM ca be found in Zelman-Femiak M and Harms GS et al.: Biotechniques 2010, Guzman A. And Knaus P. Et al. J Biol Chem. 2012 |
| 96 well plate heating insert for life cell microscopy | | PECON | M96 2000 EC | |
| Matrigel® Growth Factor Reduced (GFR) Basement Membrane Matrix | | Corning | 354230 | |
| Cytodex 3 beads | | GE Healthcare | GE17-0485-01 | |
| Scratch wound assay- Culture-Insert 2 Well in µ-Dish 35 mm | | Ibidi | #81176 | |
| µ-Slide 8 Well Glass Bottom | | Ibidi | #80827 | |
| ***Software and Algorithms*** | | | | |
| Fiji imagej | | 111 | | |
| 3D TFM analysis | TFMLAB | 108 | | Code freely available at: https://gitlab.kuleuven.be/MAtrix/Jorge/tfmlab_public |
| MATLAB | | MathWorks | | |
| Paraview | | Kitware | | |
| ZEN Blue | | Carl Zeiss | | |
| ZEN Black | | Carl Zeiss | | |
| Prism | | Graphpad | | |
| Biorender | | Biorender.com | | |

***Bmpr2b*-crispant validation**. To validate mutagenesis in *bmpr2b*-crispants, whole embryos or tail-less bodies (from crispants and control embryos) were taken for genomic DNA extraction with 50 mM NaOH and Tris pH 7.5. The following primers were used to amplify targeted regions: ZF-bmpr2b-Exon1_fwd 5′-ATGAGCTTGAACTTCGGTCGAT-3′, ZF-bmpr2b-Exon1_rev 5′- GACACCCTTTAAGACATTGCAGG-3′, ZF-bmpr2b-Exon3_fwd 5′- GTATCTCAGTGTTTCACTCCACC-3′, ZF-bmpr2b-Exon3_rev 5′- ATGACATCACAACATGCTGATACA-3′. PCR-products were sequenced with primers ZF-bmpr2b-Exon1 fwd or ZF-bmpr2b-Exon3_fwd and parts of the sequencing results are represented in Supplementary Fig. 2. Control and *bmpr2b*-crispant sequences were aligned using SnapGene 4.1.9.

### Zebrafish immunostaining

For immunostaining, zebrafish embryos were incubated for 2 h in PBS and 0.5% Triton X-100. After short washes with PBS and 0.1% Tween20 (PBST) samples were incubated for 20 min in 3% H2O2 and washed subsequently with PBST. After 2 h of blocking with 1% of BSA embryos were incubated with rabbit recombinant monoclonal CD34 antibody (EP373Y) at a 1:200 dilution and chicken anti-GFP antibody (AVES-Lab) at a 1:500 dilution overnight at 4°C. After 3 h of washing with PBST the samples were incubated over night with secondary anti-chicken-488nm antibody at a 1:200 dilution and anti-rabbit poly-HRP-conjugated secondary antibody (Tyramide SuperBoost™ Kit, Invitrogen). Samples were washed with PBST and incubated with tyramide solution at a 1:150 dilution for 40 min following the instructions of the Tyramide SuperBoost™ Kit, Invitrogen.

### Imaging and quantification of the zebrafish caudal vascular plexus

For imaging embryos were fixed with 4% paraformaldehyde overnight at 4 °C and washed and kept in PBST. Tails of the embryos were removed and mounted in SlowFade (ThermoFisher Scientific) for imaging with a LSM710 confocal microscope (Zeiss) and a ×40 objective or with a LSM880 confocal microscope (Zeiss) and a ×20 or ×40 objective. For bmpr2b-morphants analysis 90 confocal stacks (each 0.5 □m) were acquired of each tail. Of bmpr2b crispants and controls 9-12 stacks of 2.2 µm of distance were acquired. A maximum projection of the CVP was taken to count protruding cells(25-26hpf) or fenestrations (32hpf) in the ventral region or to measure the size of CVPs. AISVs rostral to the CVP were analyzed at 26hpf. For images at 32 hpf 60 stacks á 0.75 µm were acquired. Confocal images of immune-stained embryos were obtained by acquiring 8 stacks within 11.3 µm of one region of the CVP. Between 14 and 25 GFP + - cells per CVP were analyzed for CD34-presence on 2 stacks per embryo and numbers were averaged. Images were processed and analyzed with Fiji software[102]. Student′s t-test was used to test significance (p-value < 0.05).

### Reverse transcription PCR

RNA was extracted with Trizol (Sigma) from 30 embryos/sample and cDNA was synthesized from total RNA with the RevertAid H Minus First Strand cDNA Synthesis kit (ThermoFisher Scientific). Semiquantitative RT-PCR was performed using Color Taq PCR Master Mix (Roboklon) and the following primers were used: ZF-Bmpr2bE2fwd1 5′-CCAGCATGGA-GAAAATGCGT-3′; ZF-Bmpr2bE2rev1 5′-CACGATAGCCAAAGAA-CAGCG-3′; ZF-b-actinfwd 5′- TGTTTTTCCCCTCCATTGTTGG-3′; ZF-b-actinrev 5′-TTCTCCTTGATGTCACGGAC-3′.

### Western blotting

Protein lysates were separated by SDS-PAGE and subsequently transferred on PVDF membranes by Western blotting. Membranes were incubated with indicated primary antibodies overnight at 4 °C according to manufacturer's instructions. Antibodies are listed below. Chemiluminescent reactions were processed using Femto-Glo ECL reagents (PJK) and documented on a ChemiSmart5000 digital imaging system (Vilber-Lourmat). Proper loading controls (GAPDH) were examined on the same membrane.

### Immunocytochemistry

The number of $5 \times 10^4$ cells were plated on glass coverslips placed in 12-well plates. Cells were left to expand for 3–4 consecutive days to form a confluent monolayer (unless stated otherwise) before fixation with 4% paraformaldehyde (PFA) was performed. Immunofluorescence staining was performed as described in[106]. In brief, cells were permeabilized in 0.5% Triton-X-100 for 15 min at room temperature; after blocking for 1 h with a mixture of 3% w/v BSA and 5% v/v normal goat serum in PBS, cells were stained sequentially by using the indicated primary antibodies. Primary antibody binding was detected using sequential labeling with Alexa Fluor488 or Alexa594 or Alexa 647 dye conjugated secondary antibodies (Invitrogen) for 1 h at room temperature. Phalloidin-Alexa594 or Phalloidin-Alexa647 was purchased from Invitrogen and used according to manufacturer instructions. 4′,6-diamidino-2-phenylindole (DAPI; Carl Roth) was used for nuclear counterstaining. For phalloidin staining of spheroids, Phalloidin was left for 3 consecutive days under constant agitation at a 1:1000 dilution in PBS 0.1% Tween on the spheroids before washing 5 times with TBS-T buffer prior to imaging.

### Single particle tracking microscopy and trajectory analysis

Quantum dot labeled BMPR2 were tracked with MATLAB and TrackMate. The diffusion tracks on cells were visualized by color-coding TrackMate tracks by displacement. To ensure that the quantum dots were specifically bound to BMPR2 a stimulation experiment with BMP2 was performed. The mean square displacement analysis of individual BMPR2 molecules before and after stimulation with BMP2 was performed in MATLAB. Unbound quantum dots were excluded from the analysis by choosing matching tracking parameters.

### Gap closure migration assay and migration component analysis

In vitro migration was analyzed using Ibidi Culture-Insert (Ibidi). Cells (70 µl; concentration: $3.5 \times 10^5$ cells/ml) were added to Culture-Insert well and cultured for 24 h. After removal of Culture-Insert, cells were live-imaged for the durations indicated. Wound closure was recorded and gap closure are measured using ImageJ. Particle image velocimetry (PIV) analysis was performed by applying the iterative PIV (Cross correlation) plugin form ImageJ on the cell migration image series consisting in three interrogation windows with 64 pixels, 32 pixels and 16 pixels, respectively. The displacement of particles between each frame was color-coded. The migration component analysis was performed by tracking and subsequent data analysis based on 2D flow nanometry described by[57], and quantifying the velocity of the cells in y-direction and the overall diffusivity of the cells. The cells of the cell migration image series were tracked using TrackMate[107] from ImageJ. The resulting data table of tracked spots was converted to a MAT-file. Further analysis was made with a self-made MatLab (MathWorks, Natick, MA, USA) script were only tracks longer than 15 frames were included to ensure data quality. The migration speed of the cells towards the scratch ($v_y$) was quantified by calculating the velocity of the cells[20] in y-direction by polynomial curve fitting of their y-position. The directionality of the migration was quantified based on the obtained $v_y$ value and by calculating the diffusivity ($D_{xy}$) of the cells in x- and y-direction by standard deviation of the cell positions. $v_y$ and $D_{xy}$ of the two cell types were compared by a two-sided t-test each.

### Tube formation assay

BMPR2$^{wt}$ and BMPR2$^{+/-}$ cells were labeled with DiO (V-22886) and DiI (V-22885) (Vybrant™ Cell-Labeling Solutions) according to manufacturer's instructions. Briefly, cells in culture flasks were washed with PBS and corresponding dyes added to the growth medium, and cells were incubated for 20 min. In the meantime, 96 well plates were coated with growth factor reduced Matrigel® for 30 min at 37 °C. For tube formation assay, 25k labeled cells of each genotype (in total 50k cells) were mixed and seeded on Matrigel®. Tube formation was followed by taking images of the regions of interest in 30 min intervals till the end of the 6 h with an inverted epifluorescence microscope.

### 3D spheroids in Fibrin-Gel

A previously established assay[33]. Briefly, 2500 Cytodex 3 beads (GE Healthcare) were incubated with $10^6$ cells (4 h, 37 °C, and 5% CO2) and plated overnight on a 10 cm bacterial dish to remove unattached cells. Next day, the cell-covered beads were resuspended to a concentration of ~500/ml in 2,5 mg/ml fibrinogen (Sigma) solution containing 0.15 U/ml aprotinin (Sigma) and EC activation medium to a 10% concentration. Aliquots were mixed with Thrombin (Sigma; 50U/ml), distributed in 96-well plates (Grainer; 100 µl/well), and left to clot for 5–10 min. After further incubation (10 min, 37 °C, and 5% CO2), EC activation medium was added to cover the clot (Fibroblast were excluded from this experimental setting). Cytosolic staining of cells used for spheroid assay was performed using Celltracker green CMFDA or Celltracker red CMTPX (ThermoFisher Scientific) according to manufacturer's instructions.

Dual color: From each experiment, at least 4-5 positions were imaged in life cell microscopy. For the respective time points, the number of sprouting cells were counted manually. For counting, only sprouting cells with detectable color were considered. In total, at least 20 beads were taken for quantification. Percentage of green and red cells in average of 3 independent experiments were represented.

WT vs KO1: From WT and ± cells of each individual experiment, at least 6-8 beads were quantified for the sprout area. Sprout area quantification was performed using ImageJ. Initial area of each bead was measured. Then, total area of the bead was measured at the final time point and initial area of that specific bead was subtracted from the total measured area. Sprouting area of at least 30 beads were included in quantification and represented in the figure.

### Analysis of sprouting kinetics

The analysis of the sprouting kinetics of ECs in the fibrin bead assay was performed using an in-house MATLAB code on time lapse images of EC spheroids in fibrin gels. To automatically segment the spheroid boundary, we first filtered each time point spheroid brightfield images with a standard deviation filter, which enhanced the spheroid structure (which typically present heterogeneous intensity variations) while hampering the background (typically characterized by smooth intensity changes). The resultant image was binarized and processed by applying dilatation/ erosion functions and filling holes operation. The outlines of each binarized spheroid were overlapped and color-coded by time point (see Fig. 3B, top). We then computed the distance transform of the outline of the spheroid at the first timepoint (before initiation of sprouting). We computed the TC distance (see Fig. 3B, bottom-right) as the maximum value of the distance transform within the spheroid mask at a current timepoint. We computed the invasion area (see Fig. 3B, bottom-left) by measuring the difference between the area of the spheroid mask at a current timepoint and the first timepoint.

### 3D traction force microscopy

3D Traction Force Microscopy was performed by modifying a previously described method[41]. Briefly, fibrin bead assay was carried out as previously described with addition of 1% marker beads to the fibrinogen-bead suspension (Fluosphere dark red, F8807 ThermoFisher scientific). Sprouts stressed states were obtained by confocal z-stacking of sprouts of interest with a step size of 1 µm. Sprouts relaxed states were obtained by treating the sprouts with 10 µM Y-27632 (STEMCELL Technologies) for 1 h and re-imaging the sprouts as described before. Displacement vector fields and segmentation of the cell geometries was performed using the MATLAB-based open-source toolbox TFMLAB (code available at https://gitlab. kuleuven.be/MAtrix/Jorge/tfmlab_public). A detailed description of the toolbox can be found in[108].

### Displacement field analysis

The resultant displacement fields around the sprouts were further analyzed to obtain the graphs shown in Fig. 2d. Mean displacement magnitudes around a sprout were calculated by averaging the magnitude of the displacement field at the points located within a (0–5 µm) distance from the sprout surface.

Signed displacements along the direction of growth of the sprout were calculated using an in-house code as previously described in[41]. Briefly, the binarized mask of the sprout obtained from the analysis with TFMLAB, was skeletonized using MATLAB's function *bwskel*. The base and tip point of the sprout skeleton were manually selected and the principal direction of the geodesic path (defined at the skeleton's coordinates) that connects both points was taken as the growth direction of the sprout. The average displacement vectors at the points located within a (0–5 µm) distance from the sprout surface were stored taking 1 µm steps from the base to the tip of the sprout. The magnitude of these vectors (with a sign determined by their dot product with the sprout growth direction vector) at each step was plotted in the graphs shown in Fig. 2d.

### Cell shape analysis based on solidity

The tip cell solidity was calculated using an in-house MATLAB code. Tip cells ROIs were first selected manually. Then, after a contrast stretching operation, the ROIs were binarized and smoothed. The cell solidity was computed using MATLAB's function *regionprops*. This metric quantifies the ratio between the area of the cell mask and that of its convex hull (the smallest convex set containing the tip cell mask). A perfectly circular/ elliptical cell without protrusions/filopodia would have a solidity value of 1. Conversely, a lower solidity value indicates the presence of protrusions or filopodia, signifying a less regular cell shape.

### Proximity ligation assay

Transfected cells were seeded at 15.000 cells on glass cover slips in EC activation medium and left over night to adhere and spread. Subsequently, DuoLink in situ proximity ligation (Sigma Aldrich) was performed as previously described[106].

### Microscopy setups used in this study

Phase contrast and epifluorescence images were taken using an inverted fluorescence microscope AxioObserver 7 (Zeiss) with Cy2, FITC, Alexa594, and Cy5 excitation/emission filters and a ×10 Apochromat or ×63 or ×100 oil immersion objectives (Zeiss). Signals were recorded with an EMCCD camera (Zeiss). Images were processed using linear BestFit option (Zeiss). Time lapse imaging was performed on same setup for indicated time. For this samples mounted on a motorized scanning table (Maerzhaeuser, Germany) equipped with a heat- (37 °C) and CO2- (5%) controlled Life Cell Imaging chamber (Ibidi, Germany) provided a stable atmosphere. Confocal images of Supplementary Fig. 16 were acquired using inverted Leica DMi8 CEL compact semi motorized confocal scanning microscope with excitation by 405, 488, 552, and 638 nm diode lasers. Confocal data of were imaged using a 40×/1.30 HC PL APO Oil CS2 WD 0.24 mm objective, and data were recorded by photomultiplier or hybrid detector. Unless stated otherwise all other confocal data were recorded with a Zeiss LSM880 Airyscan (Zeiss) equipped with Airyscan a 40x water immersion objective in Fast acquisition mode. SPTM analysis in TIRF mode was performed with an Axiovert inverted Epi- Fluorescence microscope equipped with a 100x Plan Apochromat 1.49 NA high numerical aperture TIRF objective and a TIRF excitation Unit and AOTF and two independent laser lines with 488 nm and 561 nm excitation (VisitronSystems). A Back-illuminated EMCCD camera (Andora, Visitron Systems, Munich, Germany) was mounted on a Dual-View beam splitter with separation at 620 nm and equipped with emission filters (Chroma) to separate Quantum Dot 655 nm from Life-act GFP 488 nm emission. All images obtained were treated with ZEN Blue, ZEN Black and/or Fiji[107].

### Scanning electron microscopy

Confluent monolayers of ECs were rinsed with PBS and subjected to a primary fixation using 2.5% glutaraldehyde in PBS for 30 min. The cells were washed 3 times in PBS and subjected to a secondary fixation using 4% PFA for 20 min. The cells were then dehydrated, using serial dilutions of ethanol

(25%, 50%, 75%, 90%, and 99.9% for 5 min each). After the last dehydration step, the samples were covered with a drop of 99.9% ethanol, placed into a desiccator, and heated at 30 °C for 12 h. Prior to imaging, the samples were sputter coated with 10 nm of gold/palladium (80% gold, 20% palladium), using a BAI-TEC-SCD050 sputtering machine. Images were obtained with a Gemini-LEO-1550 scanning electron microscope (Carl Zeiss Jena, Germany), using a combination of the SE and Inlens detector set at 3 kV.

## FRET measurements

Ratiometric FRET experiments of CDC42 single chain reporter for total sensor activity were performed on a high-speed setup of Leica Microsystems (Dmi6000B stage, 63x/1.4 objective, high speed external Leica filter wheels with Leica FRET set filters[109], EL6000 light source, DFC360 FX camera, controlled by Las AF). All experiments were executed at 37 °C using a mountable incubation chamber (Ibidi) together with a cover slip holder chamber (MatTek). Ratiometric FRET measurements were performed as previously described[110]. In brief, cells transfected with single chain CDC42 biosensor were excited with Sensitized-emission FRET (seFRET) and images were recorded with the same settings for donor, acceptor, and FRET channels (8 × 8 binning, 100 ms exposure, gain 1). Images were processed with Fiji[111]. cFRET maps were calculated with PixFRET plugin[112] (threshold set to 1, Gaussian blur to 2) with a self-written macro.

## Endogenous CDC42 activity measurements

Spinning disk images were acquired with a Nikon Spinning Disk Confocal CSU-X microscope equipped with a Yokogawa CSU X-1 spinning disk unit, a 60× objective (Plan Apo VC, oil, DIC, NA = 1.4), an Andor iXon3 DU-888 Ultra EMCCD camera and the Nikon NIS elements software. GFP was imaged using a 488 nm laser line, a triple dichroic mirror (405, 488 and 561 nm) and a 521 nm emission filter. dTomato was imaged using a 561 nm laser line, a triple dichroic mirror (405, 488 and 561 nm) and a 600 nm emission filter. In brief, cells seeded in μ-Slide 8 Well Glass Bottom (Ibidi) were transfected with dTomato-WASp(CRIB) biosensor and co-transfected with either GFP or BMPR2-GFP for BMPR2[+/−] ECs the day prior to imaging. Transfected cells medium was replaced to FluoroBrite imaging medium (ThermoFisher) containing 0.1% FCS 4 h prior imaging to reduce basal activation of CDC42. Images were processed with Fiji[111]. To measure the ratio of the biosensor membrane to cytosol intensity, ROIs for the membrane and cytosol were drawn using thresholding, and the total pixel intensity value for these ROIs was measured. The ratio of the total pixel intensity value at the cell membrane over the total pixel intensity value in the whole cell was calculated and plotted as a percentage.

## Filopodia quantification and GFP fluorescence quantification

Filopodia quantification was performed using the ImageJ plugin FiloQuant in a semi-automated mode[50]. Number of filopodia was normalized to the length of the free edge at the migrating front. Total GFP fluorescence per cell was quantified using ImageJ.

## RNA sequencing data

RNA -seq dataset used to produce heatmap of CDC42 regulators expression in BMPR2[wt] and BMPR2[+/-] ECs was obtained as described previously[48].

## Statistics and reproducibility

All statistical tests were performed using GraphPad Prism version 9.3 software. All significance values were calculated by performing a two-sample two-tailed student t-test with Welch's correction was applied. For all experiments, statistical significance was assigned, with an alpha level of $p < 0.05$. Error bars all represent standard deviations in main figures and Supplementary Figs.. All data sets correspond to at least $\geq 3$ biological replicates. All attempts of replication were successful

## Reporting summary

Further information on research design is available in the Nature Portfolio Reporting Summary linked to this article.

## Data availability

The data supporting the findings of this study are available from the corresponding author upon request. Uncropped and unedited blot/gel images are shown in (Supplementary Fig. 26). Whole transcriptome RNA sequencing data presented in this study (Supplementary Fig. 24) are publicly available under the accession number GSE135312 at the National Center for Biotechnology Information advances science and health; Gene Expression Omnibus platform under the following link: https://www.ncbi.nlm.nih.gov/geo/query/acc.cgi?acc=GSE135312. DNA Sanger sequencing data for confirmation of CRISPANT zebrafish models are provided as Supplementary Data.

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

## Acknowledgements
We thank René Buschow for technical support and Paul Mendez for critical reading of the manuscript as well as all members of the Knaus Lab for their valuable input. Cartoons were generated using Biorender.com with license to P.K. and C.H. We thank Helge Ewers for providing expression vectors for BORG5 and CDC42. We would like to acknowledge the assistance of the Core Facility BioSupraMol supported by the DFG (German Research Foundation). The authors gratefully acknowledge the MPI-MG for granting access to microscopes within the framework of the IMPRS-BAC. C.H., P.K. and H.V.O. would like to thank Research Foundation—Flanders (FWO, grant W001420N) and An Zwijsen for stimulating this cooperation. P.K. was supported by the DFG (SFB1444) and the Morbus Osler Stiftung. J.B-F. gratefully acknowledges the support from FWO (junior postdoctoral grant 1259223 N). H.V.O. was supported by iBOF/21/083 C; G.H. was supported by a Howard Hughes Medical Institute grant and funding from both Wilkes University (USA) and the University of Mainz Medical Center, Y.K. and S.B. acknowledge funding by the DFG within project BL1514/1. S.A.-S. was generously supported by SFB958, Deutsche Forschungsgemeinschaft (DFG) projects SE2016/7-3, SE2016/10-1, SE2016/13-1, and the Leducq Transatlantic Network of Excellence "21CVD03 - ReVAMP". M.I. receives a stipend from Berlin School for Integrative Oncology (BSIO). N.H. receives a stipend from Sonnenfeld Stiftung. M.B. receives a stipend by the International Max Planck Research School for Biology and Computation.

## Author contributions
C.H. and P.K. conceptualized the study, C.H. performed experiments dedicated to cytoskeleton, migration, adhesion, SPTM and CDC42. M.B. supported C.H. for several experiments, quantified critical data and optimized and performed all 3D sprouting angiogenesis assays and 3D TFM experiments including analysis, and PI3K and CDC42 sensor experiments. T.M. assisted in multichannel confocal microscopy for 3D TFM. Y.K performed analysis of SPTM data and together with S.B. conceptualized and performed the migration component analysis. G.H. helped in performing SPTM experiments and provided microscopy hardware and expertise. B.K. supported in analysis of FRET data. S.S. and J.M. performed and analyzed experiments in zebrafish. M.I. helped in quantifying the 3D mosaic sprouting data. N.H. performed tube formation assay. M.C., J.B. and H.V.O. advised on 3D TFM experiments, J.B. and H.V.O. shared critical software and supported in data analysis and interpretation. O.R. supported with the biosensor experiments. C.H. and M.B. assembled the figures. C.H, M.B. and P.K. wrote the manuscript. All authors read and commented on the manuscript.

## Funding

## Competing interests
The authors declare no competing interests.
