## [Transparent Peer Review file · Communications Biology]

Endothelial tip-cell position, filopodia formation and biomechanics require BMPR2 expression and signaling

Corresponding Author: Professor Petra Knaus

Version 0:

Reviewer comments:

Reviewer #1

(Remarks to the Author)

The paper by Hiepen et al addresses the importance of BMPR2 in endothelial tip cell biomechanics during angiogenic sprouting. Ablation of this protein impairs endothelial cell sprouting and filopodia formation due to a failure of cells to acquire tip cell positions. The paper is novel and interesting, but some of the interpretations require additional data to support the conclusions made.

Specific Comments.

Major points:

Figure 4b: The quality of the AKT blots is poor and should be repeated. It would be preferable to use both Ser473 and Thr308 phosphorylation site antibodies and total AKT for completion of analysis and to confirm interpretation of data. It is surprising that LY294002 treatment affects WT cells more than BMPR2-depleted cells. Is this consistent across experiments, as there is no quantification of data, or mention that this blot is representative of several replicates. There is also no LY294002 treatment of the siRNA knockdown HUVECs. Could the authors clarify the conditions the cells are exposed to? From the methods section, it appears these cells are exposed to "LY294002 or DMSO in 20% serum for 60 minutes prior to stimulation". Can the authors confirm whether the cells were further stimulated? The AKT phosphorylation signal is acute, responding to transient generation of PtdIns(3,4,5)P₃. For completion, the authors should also present data showing cells that have been serum-starved and then exposed to an acute proangiogenic signal to activate the PI3K signalling pathway in endothelial cells. The figure legend should include the concentrations and exposure times of the inhibitors used. In order to verify that this pathway is PI3K-dependent and prevent overinterpretation of data, rescue experiments should be performed using constitutively active PI3K or AKT constructs.

Figure 4c: The localisation of both CDC42 and BMPR2 appear to be quite punctate throughout the entire cell. Could the robustness of the localisation data be addressed in the manuscript?

Figure 1 supplement 7: Authors may need to address the issue of the effect that high expression of BMPR2 may have on the cell in terms of off-target or dominant-negative effects that may mask any valid interpretation of results here. Were DNA concentrations used other than the two shown here?

Line 229: It would be useful to stain the cells for cell-cell adhesion markers to test the claim that differences in migration are due to changes in cell-cell adhesion.

Figure 1 supplement 8: There is still strong phosphorylation of SMAD in BMPR2 WT ECs upon exposure to LDN, which makes it difficult to support the conclusion drawn on line 178. Could the authors address this?

Figure 2 supplement 1: The BMPR2 cells appear to spread more than the WT cells. This experiment could be repeated with increased cell confluence, as the WT cells appear to be subconfluent, which will affect the ability of the cells to form cell-cell junctions and could address whether this morphology is due to increased cell spread or increased cell-cell junction formation. The use of cell-cell adhesion markers will also help to address this question.

Minor Points:

Figures 1e and 1f: Please show a larger field of view, so that cell morphology can be assessed across more than one cell.

Figure 1 supplement 4: Provide representative images.

Figure 1 supplement 5: Please show a larger field of view, so that cell morphology can be assessed across more than one cell.

Figure 1 supplement 6: Provide representative images.

Figure 1 supplement 8: Provide total SMAD blots for full interpretation of data.

Figure 1 supplement 8: Provide representative images that correspond with the graph. Can the authors clarify the reason for the different replicate numbers shown between each condition?

Figure 1 supplement 8: The figure legend should include the exposure times of the inhibitors used.

Figure 2c: Please show a larger field of view, so that localisation can be assessed across more than one cell.

Figure 2 supplement 2: Provide representative images for all substrates. Show a larger field of view so that focal adhesion formation can be assessed in more than one cell.

Figure 4a: Please show a larger field of view with cells with varied expression of CDC42. Cells with very high expression of CDC42 may have off-target or dominant-negative effects and this needs to be addressed.

Reviewer #2

(Remarks to the Author)

The manuscript by Hiepen et al. addresses the potential role of the type II receptor, BMPR2, in the control of endothelial morphogenic sprouting and tip cell development in vivo in Zebrafish and using in vitro assays. They conclude that BMPR2 plays an important role in filopodia formation, a process that is known to involve the small GTPase, Cdc42 and its effectors. Their Zebrafish experiments involved only morpholino knockdown of BMPR2. They did not disrupt the gene, which makes the conclusions in vivo, unclear, and incomplete. There are major issues with the in vitro assay approaches also, and there are other significant deficiencies in evaluating how BMPR2 exerts its potential influence on tip cell behavior, such as filopodia formation. For example, which BMP ligand is involved in driving the effect that they are investigating. Also, which type 1 BMP receptor (or receptors) is involved in the effect that they are investigating? These key questions are not at all addressed.

Specific Comments

1. There are numerous technical and experimental issues with the presented studies. These deficiencies make the interpretation of the conclusions difficult and overall, the conclusions are very unclear. The use of only a Zebrafish morpholino approach is not sufficient to answer the question that they are asking. A gene knockout is also necessary. This is particularly critical because the in vitro assays are also very problematic and not very convincing. The media conditions in sprouting assays are not defined in that they utilized the Hughes fibrin model, but don't specify the media conditions or whether fibroblasts are added to the system. Scratch wound assays are not at all relevant in this context. Tip cells are defined by a 3D matrix environment, not 2D migratory behavior. The use of the Matrigel as a measure of EC lumen formation is inadequate, as ECs do not form lumens on Matrigel surfaces, and thus, do not make tubes. At best, it is an assay of cellular cord assembly, but many cell types other than ECs can form cords on this surface. Another key point is that in vitro morphogenic assays that assess tip cell behavior must also have convincing lumen formation, which is not demonstrated in their data set either. If every cell is invading individually, there is no way to assess tip cells vs. stalk cells (i.e. lumen forming cells) in these assays.

2. There is no consideration of the role of growth factors that bind BMPR2 or its associated type 1 receptors in this work. BMPR2 has many potential Alk receptors that could be relevant here, and possibly other binding partners that could affect these results. Also, since no information is presented on relevant BMP ligands, it is difficult for this reviewer to interpret the data regarding reducing expression of BMPR2. In theory, this could also affect the binding of other Alks to other type 2 receptors such as endoglin, which could then affect the results. In addition, the Hughes fibrin assay system as originally published has brain extract, various added growth factors, fetal calf serum, fibroblasts present, so it the likelihood is that many BMPs could be present or molecules that trap or affect the activity of BMPs. This could certainly be an issue on whether EC tip cells have filopodia. Growth factor signaling is certainly important for this, but no clear information is provided on which growth factors are controlling this response in this study.

3. Cdc42 has many effectors, so it is unclear whether other effectors including Pak kinases, IQGAPs, MRCKs etc. could be involved in the Cdc42-dependent aspects of the tip cells. Were these key Cdc42 effectors screened? Although Borg5 may play a role, it is likely that other Cdc42 effectors would be involved in this response.

Reviewer #3

(Remarks to the Author)

This is an important study by Hiepen et al. that establishes a decisive role for BMPR2 in filopodia formation during sprouting angiogenesis. Several overlapping cellular models are used to demonstrate the biochemical and biomechanical properties controlled by BMPR2 to provide directional migration especially for leading tip cells. The presented data are of high quality and rigorously analyzed. There are a few minor concerns and comments, which when addressed, will likely further strengthen the conceptual impact of the study.

1. While the requirement of BMPR2 in filopodia formation and directional migration is clear, the molecular mechanism linking BMPR2 to BORG5 and regulation of CDC42 could be further interrogated. A rescue experiment may be informative using BMPR2-kinase dead/deficient point mutants. Recent structural studies of the BMPR2 intracellular domain show that some of these point mutants associated with vascular disorders may fold and traffic relatively normally.
2. It is unclear whether the BMPR2-dependent filopodia formation and sprouting are primarily occurring in vein-associated ECs (caudal vein, HUVECs)? How about the intersegmental vessels in zebrafish embryos or microvascular ECs?

Minor points:

1. Fig.1b- Precisely how many embryos and ROIs are represented in the data?
2. Please provide greater detail on how BMPR2[±] ECs were derived. Were these CRISPR-edited cells derived from single cell clones or pooled? Compared to control, the haploinsufficient ECs appear more like a knockout.
3. Co-staining with hallmark tip cell markers (e.g. CD34 or ESM1) in some of the experiments such as the leader cells in the scratch assay or spheroid sprouting would be helpful.

Version 1:

Reviewer comments:

Reviewer #1

(Remarks to the Author)

The authors have done an excellent job addressing our questions, comments and recommendations. The additional experimental data, which encompasses multiple approaches to address their biological questions, strengthens and improves the manuscript and its scientific interpretations regarding a central role for BMPR2 in endothelial tip cell filopodia formation.

Reviewer #2

(Remarks to the Author)

The authors have addressed my concerns and have added some convincing new data to support their conclusions.

Reviewer #3

(Remarks to the Author)

The authors have adequately addressed the concerns through new data and discussions. Their efforts to address other reviewers' critique through new experimental data are also noted.

Dear Reviewers,

Thank you for the opportunity to revise our manuscript entitled “Endothelial tip-cell position, filopodia formation and biomechanics require BMPR2 expression and signaling”. We have worked diligently to address all reviewer’s comments, many of which prompted us to conduct new and rigorous experiments.

Notably, we have implemented genetic targeting of the BMPR2 gene in zebrafish to complement our earlier transient targeting studies and underscore the critical role of its disruption in filopodia formation. Furthermore, we conducted a thorough set of signaling experiments using small molecule inhibitors and activators to elucidate the roles of specific BMPs, type I receptors, and BMPR2 kinase activity within the context of our proposed mechanism. In addition, we established an endogenous CDC42 activity biosensor to validate BMPR2's role in CDC42 activation at the endothelial plasma membrane, crucial for filopodia formation. This was supported by Oliver Rocks, who is placed as co-author to the revised manuscript now. Below we will place a list of all new figures and those, which are now positioned differently, to make the progress more visible.

Reviewer #1 (Remarks to the Author):

The paper by Hiepen et al addresses the importance of BMPR2 in endothelial tip cell biomechanics during angiogenic sprouting. Ablation of this protein impairs endothelial cell sprouting and filopodia formation due to a failure of cells to acquire tip cell positions. The paper is novel and interesting, but some of the interpretations require additional data to support the conclusions made.

Specific Comments.

Major points:

Figure 4b: The quality of the AKT blots is poor and should be repeated. It would be preferable to use both Ser473 and Thr308 phosphorylation site antibodies and total AKT for completion of analysis and to confirm interpretation of data.

1.1 Answer: We performed experiments as required. Original Figure 4b is now new Figure 4e. To better characterize AKT phosphorylation and PI3K activity in our cell models, we have repeated our western blot analysis by targeting both phospho-AKT Ser473 and phospho-AKT Thr308 sites, as well as total amount of AKT. A representative blot can be found as a new figure 4e, with quantification of phospho-AKT Ser473 and phospho-AKT Thr308 under DMSO, LY, or UCL-TRO-1938 treatment as new Figure 4e and Figure 4-figure supplement 1. These results were added and discussed in main text (line 462-69).

It is surprising that LY294002 treatment affects WT cells more than BMPR2-depleted cells. Is this consistent across experiments, as there is no quantification of data, or mention that this blot is representative of several replicates.

1.2 Answer: We have performed more experiments and quantifications to address this concern. Results show, that no differences between LY treatments were observed between wt and BMPR2-depleted cells. We have quantified AKT phosphorylation in our cell models upon PI3K activation and inhibition in several

experimental repeats to show the robustness of PI3K inhibition or activation upon LY treatment or UCL-TRO-1938 treatment, respectively, in both wild-type and BMPR2 depleted ECs. These quantifications are presented in new Figure 4- figure supplement 1. These results were added and discussed in main text (line 465-69)

There is also no LY294002 treatment of the siRNA knockdown HUVECs.

1.3 Answer: We have decided to remove the corresponding dataset from the revised manuscript. By performing further experiments, we have realized that double treatment of cells with siRNA and PI3K inhibitor LY leads to reduced cell viability. Instead, we have performed the LY treatment in our genetic model revealing that the inhibitor works robustly in our BMPR2-depleted cells and that filopodia formation is inhibited.

Could the authors clarify the conditions the cells are exposed to? From the methods section, it appears these cells are exposed to “LY294002 or DMSO in 20% serum for 60 minutes prior to stimulation”.

1.4 Answer: We thank the reviewer for careful reading of our manuscript. To clarify the conditions that cells were exposed to, we have rephrased the material and methods as such: “Pharmacological inhibitors LDN-193189 (5 μ M), LY294002 (10 μ M) and small molecule activator UCL-TRO-1938 (10 μ M) were added in EC activation medium containing 20 % Serum and pro-angiogenic growth factors for 60 min prior to lysis.” (lines 643ff)

Can the authors confirm whether the cells were further stimulated? The AKT phosphorylation signal is acute, responding to transient generation of PtdIns(3,4,5)P₃. For completion, the authors should also present data showing cells that have been serum-starved and then exposed to an acute proangiogenic signal to activate the PI3K signalling pathway in endothelial cells.

1.5 Answer: To answer the concern of PI3K transient activation upon a pro-angiogenic cue, we have performed pAKT level analysis and filopodia quantification in wound healing assays of BMPR2^{wt} ECs or BMPR2^{+/-} ECs under BMP6 stimulation. Previous work by us has shown BMP6 to be a potent pro-angiogenic ligand (Benn et al., *FASEB J.* 2017) and to be a fitting candidate trigger for the mechanism described in this current study. The results of these assays are presented in Rebuttal Figure 1. Our results show a trend indicating that BMP6 could marginally induce filopodia formation in BMPR2^{wt} ECs but not in BMPR2^{+/-} ECs (Rebuttal Figure 1a, b). However, western blot analysis of pAKT-Ser473 and pAKT-Thr308 upon BMP6 stimulation of both BMPR2^{wt} ECs and BMPR2^{+/-} ECs revealed no change compared to starved only ECs (Rebuttal Figure 1c, d). Therefore, although BMP6 seems to be able to promote filopodia formation in BMPR2^{wt} ECs, the lack of robust PI3K activation downstream of BMP6 stimulation seem to indicate that another BMP ligand is responsible for the strong BMPR2-dependent filopodia formation observed when ECs were culture in EC activation medium. Further experiments would be required to specify which BMP ligand would be responsible for this effect. As the results of these experiments were inconclusive in regard of the BMPR2-dependent PI3K-induced filopodia formation, they were not added to our manuscript.

Rebuttal Figure 1. PI3K activation and filopodia formation in response to BMP6 stimulation: a Representative images of cell edge of BMPR2wt ECs or BMPR2^{+/-} ECs cultured in EC starvation medium for 4 hours prior insert removal and stimulated with 10nM BMP6 upon insert removal for 3h before fixation. **b** Quantifications of the number of filopodia per 100µm cell edge of BMPR2wt ECs or BMPR2^{+/-} ECs cultured in EC starvation medium for 4 hours prior insert removal and stimulated with 10nM BMP6 upon insert removal for 3h before fixation. *p<0.05, n.s: non significant. **c** immunoblots against pAKT-Ser473 and pAKT-Thr308 from BMPR2wt ECs or BMPR2^{+/-} ECs cultured in EC starvation medium for 4 hours prior stimulation with 10nM BMP6 for 60mn before lysis. **d** Quantification of immunoblots against pAKT-Ser473 and pAKT-Thr308 from BMPR2wt ECs or BMPR2^{+/-} ECs cultured in EC starvation medium for 4 hours prior stimulation with 10nM BMP6 for 60mn before lysis. pAKT signal was normalized to total AKT for each condition. n.s: non significant.

The figure legend should include the concentrations and exposure times of the inhibitors used.

1.6 Answer: We thank the reviewer for careful reading. We have included the LY294002 concentration and exposure time used in the legend of figure 4 and Figure 4 – figure supplement 1.

In order to verify that this pathway is PI3K-dependent and prevent overinterpretation of data, rescue experiments should be performed using constitutively active PI3K or AKT constructs.

1.7 Answer: We agree that our PI3K-data are not allowing to fully describe BMPR2-dependent filopodia formation on a mechanistic level. Our interpretation is, that PI3K signalling could be one of several pathways downstream of BMPR2 yet to be discovered. However, rescue of PI3K signalling in BMPR2-deficient cells using the small molecule activator of PI3K catalytic subunit UCL-TRO-1938 also rescued filopodia formation (Figure 4f), while LY treatment blunts filopodia formation in wt cells (Figure 4f). Additionally, pAkt seems reduced in mutant cells (Figure 4e and Figure 4 – figure supplement 1). Together with our new data, highlighting the role of CDC42 in this process and PI3K being a major upregulator of CDC42, we suggest that PI3K signalling is one out of several potential players involved in the here reported

phenotype. However, we agree that overinterpretation should be avoided and therefore we have attenuated our emphasize on the role of PI3K signalling.

Figure 4c: The localization of both CDC42 and BMPR2 appear to be quite punctate throughout the entire cell. Could the robustness of the localization data be addressed in the manuscript?

1.8 Answer: We thank the reviewer for their input. Original Figure 4c is now Figure 4a. To better resolve the colocalization of BMPR2 and CDC42 in a more precise manner, we have performed co-expression of BMPR2-GFP and of an active endogenous CDC42 localization biosensor (new Figure 4g), namely dTomato-WASp(CRIB) in BMPR2^{+/-} ECs. The results of this assay could show the clear colocalization of active CDC42 and BMPR2 -GFP at the plasma membrane of transfected BMPR2^{+/-} ECs and clearly indicated the necessity of membrane BMPR2 for CDC42 recruitment to the PM and its activity. These results are shown in the new Figure 4g and are discussed in the main text (line 490-501) and the corresponding materials and method were added to the corresponding section.

Figure 1 supplement 7: Authors may need to address the issue of the effect that high expression of BMPR2 may have on the cell in terms of off-target or dominant-negative effects that may mask any valid interpretation of results here. Were DNA concentrations used other than the two shown here?

1.9 Answer: We thank the reviewer for the input. Original figure 1 – figure supplement 7 is now Figure 1 – Supplementary 10. To better assess the correlation between level of BMPR2-GFP overexpression in BMPR2^{wt} ECs and resulting filopodia, we have quantified the number of filopodia for several cells showing different level of GFP intensities when imaged with the same exposure time after BMPR2-GFP transfection using a single amount of DNA (1,4µg of DNA transfected via Lipofectamine²⁰⁰⁰ on cells in 70-80% confluent well of a 12 well plate at the time of transfection). The resulting graph has been added to Figure 1 – Supplementary 10 and the results were discussed in main text (lines 190-193)

Our Data suggests a strong correlation between BMPR2-GFP expression and filopodia numbers in BMPR2^{wt} ECs up to saturation conditions.

Line 229: It would be useful to stain the cells for cell-cell adhesion markers to test the claim that differences in migration are due to changes in cell-cell adhesion.

1.10 Answer: We have performed immunofluorescence staining of both wildtype and BMPR2 depleted EC junctions via targeting of adherens junction located Beta-Catenin. These stainings were indicative of broader, sturdier junctions in the absence of BMPR2 in ECs. These new stainings are presented as new Figure 2c, and resulting findings are discussed in the text (lines 287-290). Reference to the anti- Beta-Catenin antibody were added to the Key resource table.

Figure 1 supplement 8: There is still strong phosphorylation of SMAD in BMPR2 WT ECs upon exposure to LDN, which makes it difficult to support the conclusion drawn on line 178. Could the authors address this?

1.11 Answer: Original figure 1 – figure supplement 8 is now split into new Figure 1 – Supplementary 11 and new Figure 1 – Supplementary 12. We have repeated LDN treatment of both BMPR2^{wt} and BMPR2^{+/-} ECs and HUVEC treated with scrambled or BMPR2-targeting siRNA (n = 3 biological replicates) and quantified pSmad1/5 level upon LDN inhibition. Our data confirm the robustness of pSmad1/5 depletion in all cell types upon treatment. Quantification is presented in new Figure 1- figure supplement 11 and is

discussed in the main text (lines 208-214). Therefore, we are convinced that BMPR2-dependent filopodia formation is merely BMP-SMAD independent.

Figure 2 supplement 1: The BMPR2 cells appear to spread more than the WT cells. This experiment could be repeated with increased cell confluence, as the WT cells appear to be subconfluent, which will affect the ability of the cells to form cell-cell junctions and could address whether this morphology is due to increased cell spread or increased cell-cell junction formation. The use of cell-cell adhesion markers will also help to address this question.

1.12 Answer: We thank the reviewer for the input. We have replaced Figure 2 supplement 1 to show ECs seeded at comparable densities to better assess the difference in cell spreading. Furthermore, cell-cell adhesion marker Beta-Catenin was stained via immunofluorescence in both BMPR2^{wt} and BMPR2^{+/-} ECs. These new stainings are presented as new figure 2c, and resulting findings are described in the text (lines 283-290).

Minor Points:

Figures 1e and 1f: Please show a larger field of view, so that cell morphology can be assessed across more than one cell.

1.13 Answer: We have replaced both Figures 1e and 1f to show a larger field of view so that cell morphology can be assessed across several ECs.

Figure 1 supplement 4: Provide representative images.

1.14 Answer: Original figure 1 – figure supplement 4 is now Figure 1 – Supplementary 7. We have added representative images showing typical filopodia protrusions for all conditions for which filopodia length was quantified and edited the figure legend accordingly. These new data are introduced in Figure 1 - Supplement 7.

Figure 1 supplement 5: Please show a larger field of view, so that cell morphology can be assessed across more than one cell.

1.15 Answer: Original figure 1 – figure supplement 5 is now Figure 1 – Supplementary 8. We have shown a larger field of view so that cell morphology can be assessed across several ECs. These new data are introduced in Figure 1 - Supplement 8

Figure 1 supplement 6: Provide representative images.

1.16 Answer: Original figure 1 – figure supplement 6 is now Figure 1 – Supplementary 9. We have added representative images showing typical filopodia protrusions for BMPR2^{wt} ECs transfected with either GFP or BMPR2-GFP as Figure 1 - Supplement 9

Figure 1 supplement 8: Provide total SMAD blots for full interpretation of data.

1.17 Answer: Original figure 1 – figure supplement 8 is now split into new Figure 1 – Supplementary 11 and new Figure 1 – Supplementary 12. We have added the total SMAD1 immunoblot to complement this figure. This immunoblot can be found in the new Figure 1 - Supplement 11

Figure 1 supplement 8: Provide representative images that correspond with the graph. Can the authors clarify the reason for the different replicate numbers shown between each condition?

1.18 Answer: Original figure 1 – figure supplement 8 is now split into new Figure 1 – Supplementary 11 and new Figure 1 – Supplementary 12. We have added representative images showing typical filopodia protrusions for BMPR2^{+/-} ECs overexpressing GFP or BMPR2-GFP and having been treated with DMSO or 5µM LDN-193189 in Figure 1 - Supplement 12. The low replicates number for the LDN treated BMPR2-GFP transfected ECs can be explained by the stress that they undergo upon seeding on glass coverslips followed by both lipofection in serum-free medium and LDN treatment before fixation. This resulted in cell death making the imaging of positively transfected cells challenging. However, ECs transfected with BMPR2-short form and also treated with LDN showed better survival and allowed for increased replicate numbers which allowed us to make the point that BMPR2, albeit without its characteristic tail, was still able to rescue filopodia formation even when BMP type 1 receptors were inhibited.

Figure 1 supplement 8: The figure legend should include the exposure times of the inhibitors used.

1.19 Answer: Original figure 1 – figure supplement 8 is now split into new Figure 1 – Supplementary 11 and new Figure 1 – Supplementary 12. We have included the LDN-193189 concentration and exposure time used in the corresponding figure legend.

Figure 2c: Please show a larger field of view, so that localization can be assessed across more than one cell.

1.20 Answer: Original Figure 2c is now Figure 2d and was modified to show a larger field of view so that cell morphology can be assessed across several ECs for both BMPR2^{wt} and BMPR2^{+/-} ECs.

Figure 2 supplement 2: Provide representative images for all substrates. Show a larger field of view so that focal adhesion formation can be assessed in more than one cell.

1.21 Answer: We have replaced Figure 2 - Supplement 2 to include representative images for all the different substrates that were used for this assay and for following Focal Adhesion circularity quantifications.

Figure 4a: Please show a larger field of view with cells with varied expression of CDC42. Cells with very high expression of CDC42 may have off-target or dominant-negative effects and this needs to be addressed.

1.22 Answer: We thank the reviewer for the comment. Original figure 4a is now figure 4c and was replaced to show a larger field of view so that cell morphology and CDC42 expression can be assessed across several ECs.

Reviewer #2 (Remarks to the Author):

The manuscript by Hiepen et al. addresses the potential role of the type II receptor, BMPR2, in the control of endothelial morphogenic sprouting and tip cell development in vivo in Zebrafish and using in vitro assays. They conclude that BMPR2 plays an important role in filopodia formation, a process that is known to involve the small GTPase, Cdc42 and its effectors. Their Zebrafish experiments involved only morpholino knockdown of BMPR2. They did not disrupt the gene, which makes the conclusions in vivo, unclear, and incomplete. There are major issues with the in vitro assay approaches also, and there are other significant deficiencies in evaluating how BMPR2 exerts its potential influence on tip cell behavior, such as filopodia formation. For example, which BMP ligand is involved in driving the effect that they are investigating. Also, which type 1 BMP receptor (or receptors) is involved in the effect that they are investigating? These key questions are not at all addressed.

We thank the reviewer for these critical comments.

We agree that a genetic Knock out of BMPR2 in the zebrafish model is essential to understand the complex effect of the absence of BMPR2 on vasculature development and sprouting angiogenesis in our in vivo model. We therefore performed a Crispant knock out deletion of *Bmpr2b* to better understand the phenotype described in our original submission.

However, we mostly disagree with the statement that the in vitro assays presented in our study are inadequate and are confident about the relevance of their use to better understand BMPR2-driven filopodia formation and sprouting angiogenesis in the context of our manuscript.

Finally, we fully acknowledge the importance of interrogating which further BMP signaling components — whether BMP ligands or BMP type I receptors—might be at play in the mechanism hereby described.

Each of those points are addressed in the following answer to the reviewer.

Specific Comments

1. There are numerous technical and experimental issues with the presented studies. These deficiencies make the interpretation of the conclusions difficult and overall, the conclusions are very unclear. The use of only a Zebrafish morpholino approach is not sufficient to answer the question that they are asking. A gene knockout is also necessary.

2.1 Answer: We have generated *bmpr2b*-crispant zebrafish to address the reviewer concern and to complement the *bmpr2b*-morphant zebrafish we introduced in our original submission (Figure 1a). We applied a crispant approach (Kroll et al. 2021) targeting *bmpr2b* exon 1 and 3 to disrupt the gene in zebrafish in F0. We show this by sequencing the targeted region (Figure 1 - supplementary figure 2). Similarly to the previously submitted *bmpr2b*-morphant zebrafish data, those gene mutations disturbed endothelial sprouting in crispant embryos CVPs (Figure 1a, b). We therefore confirmed that interfering at different loci on *bmpr2b* via transient RNA approach

(morphants) and robust genetic approach (crisprants) to disrupt *Bmpr2b* function, perturbs endothelial sprouting in the CVP. We can thus conclude that *Bmpr2b* is required for sprout formation in the CVP. These new findings are discussed in the results section (lines 146-155).

This is particularly critical because the in vitro assays are also very problematic and not very convincing. The media conditions in sprouting assays are not defined in that they utilized the Hughes fibrin model, but don't specify the media conditions or whether fibroblasts are added to the system.

2.2 Answer: We have clearly described under material and methods the conditions, under which the 3D sprouting assays in fibrin were performed, recapitulating previous work, in which the same assay with only pro-angiogenic medium was used to assess sprouting angiogenesis and TC/SC formation (1). All sprouting assays were performed in the absence of fibroblasts, as those are dispensable to promote robust sprouting angiogenesis if growth factors are provided to the spheroids via their medium. We could show that our EC activation medium could promote steady and durable sprouting angiogenesis of our cells, demonstrating that it contains potent pro-angiogenic factors sufficient to promote angiogenesis in the absence of fibroblasts.

Scratch wound assays are not at all relevant in this context. Tip cells are defined by a 3D matrix environment, not 2D migratory behavior.

2.3 Answer: We fully agree that wound healing assays are not adequate to recapitulate tip cell formation due to the lack of a 3D environment around the ECs. However, we described the use of this assay in our manuscript as relevant to assess EC characteristics such as filopodia formation and migration upon EC polarization, which are TC-related features. Therefore, we were careful not to label the leader cells as Tip Cells in our results and to reassess those same characteristics in 3D sprouting assays later in the manuscript (Figure 3). Our 2D findings for leader cells (less filopodia and migration upon *BMPR2* depletion) were recapitulated by the tip cells formed in our 3D sprouting assays (Figure 3a, b and c)

The use of the Matrigel as a measure of EC lumen formation is inadequate, as ECs do not form lumens on Matrigel surfaces, and thus, do not make tubes. At best, it is an assay of cellular cord assembly, but many cell types other than ECs can form cords on this surface.

2.4 Answer: We very much agree with the reviewer the tube formation assay is not a fitting method to investigate endothelial lumen formation in vitro. However, we never intended on investigating the role of *BMPR2* in vascular lumen establishment as we are here examining very early *BMPR2*-dependent molecular mechanism participating in tip cell formation, thus much preceding lumen formation. In our study, we make use of this assay in a *BMPR2*^{wt} vs *BMPR2*^{+/-} mosaic set-up to better understand the propension of one EC type over the other to occupy certain positions in the tube-network formed, thus highlighting differences in migratory and adhesive properties upon *BMPR2* deficiency. The reasoning behind the use of this assay is also explained in our manuscript (line 313ff).

Another key point is that in vitro morphogenic assays that assess tip cell behavior must also have convincing lumen formation, which is not demonstrated in their data set either. If every cell is invading individually, there is no way to assess tip cells vs. stalk cells (i.e. lumen forming cells) in these assays.

2.5 Answer: We do not agree with the reviewer that tip cell specialization is dependent on lumen establishment. Previous literature describe sprouting angiogenesis as a partly sequential process where Tip cell specialization, polarization and invasion and SC formation and proliferation shortly precede the formation of a lumen (2). After which TC and SC maintenance and lumen formation happen concomitantly during sprouting angiogenesis. We therefore believe that this experimental set-up recapitulates the steps required to the formation of TC and following SCs even without the formation of sprout lumens, and allows to investigate the role of BMPR2 in the molecular mechanisms regulating the early steps of TC specialization.

References

1. Wimmer R, Cseh B, Maier B, Scherrer K, Baccharini M. Angiogenic sprouting requires the fine tuning of endothelial cell cohesion by the Raf-1/Rok-alpha complex. *Dev Cell.* 2012;22(1):158-71.
2. Lizama CO, Zovein AC. Polarizing pathways: balancing endothelial polarity, permeability, and lumen formation. *Exp Cell Res.* 2013;319(9):1247-54.

2. There is no consideration of the role of growth factors that bind BMPR2 or its associated type 1 receptors in this work. BMPR2 has many potential Alk receptors that could be relevant here, and possibly other binding partners that could affect these results.

2.6 Answer: It is important to note that BMPR2 is a vastly promiscuous BMP receptor which might be involved in signaling complexes including numerous BMP type 1 receptors as well as a variety of BMP ligands, while also being able to signal via BMP/SMAD-independent routes. Nonetheless, to better evaluate the specific role of BMP Type 1 receptors and co-receptors in endothelial filopodia formation and their ability to rescue this process in the absence of BMPR2, we have overexpressed Alk1-GFP, Alk2-GFP, Alk3-GFP and Endoglin-GFP in BMPR2^{+/-} ECs and quantified subsequent filopodia formation. These new data are presented in the new Figure 1 - Supplement 15 and discussed in main text (lines 221-24), and corresponding constructs were added to the Key resource table. Our data suggest that none of the type 1 receptors or Endoglin are able to rescue endothelial filopodia formation in the absence of BMPR2, therefore highlighting the unique role carried out by BMPR2 in this process.

Also, since no information is presented on relevant BMP ligands, it is difficult for this reviewer to interpret the data regarding reducing expression of BMPR2. In theory, this could also affect the binding of other Alks to other type 2 receptors such as endoglin, which could then affect the results. In addition, the Hughes fibrin assay system as originally published has brain extract,

various added growth factors, fetal calf serum, fibroblasts present, so it the likelihood is that many BMPs could be present or molecules that trap or affect the activity of BMPs. This could certainly be an issue on whether EC tip cells have filopodia. Growth factor signaling is certainly important for this, but no clear information is provided on which growth factors are controlling this response in this study.

2.7 Answer: We agree with the reviewer that better understanding whether and which BMP ligands are at play in BMPR2-dependent endothelial filopodia formation is of interest, without excluding the possibility that this mechanism might be independent of BMP ligands. To answer this concern, we have performed filopodia quantification in wound healing assays and pAKT level analysis to assess PI3K activation under BMP6 stimulation, as previous work by us have shown BMP6 to be a potent pro-angiogenic ligand (Benn et al., *FASEB J.* 2017). BMP6 could be a fitting candidate acting upstream of BMPR2 for filopodia formation. Our results show a trend indicating that BMP6 could slightly induce filopodia formation in BMPR2^{wt} ECs but not in BMPR2^{+/-} ECs. However, western blot analysis of pAKT-Ser473 and pAKT-Thr308 upon BMP6 stimulation of both BMPR2^{wt} ECs and BMPR2^{+/-} ECs revealed no change compared to starved only ECs. Therefore, although BMP6 was marginally able to promote filopodia formation in BMPR2^{wt} ECs, the lack of robust PI3K activation by BMP6 stimulation seems to indicate that BMP6 is not responsible for the strong BMPR2-dependent activation of PI3K upstream of filopodia formation observed in ECs cultured in EC activation medium. The results of these assays are presented in Rebuttal figure 1 which was addressed to reviewer #1 in the **answer 1.5** earlier in this letter.

The results of these experiments were thus inconclusive in regard of the BMPR2-dependent PI3K-induced filopodia formation and further investigating whether BMPs and which ones are involved in the process hereby described would need further experiments beyond the scope of this project. Therefore, they were not added to our manuscript.

3. Cdc42 has many effectors, so it is unclear whether other effectors including Pak kinases, IQGAPs, MRCKs etc. could be involved in the Cdc42-dependent aspects of the tip cells. Were these key Cdc42 effectors screened? Although Borg5 may play a role, it is likely that other Cdc42 effectors would be involved in this response.

2.8 Answer: We appreciate the reviewer's inquiry regarding the potential involvement of specific CDC42 effectors in CDC42-dependent tip cell characteristics. We have performed RNA sequencing analysis of BMPR2^{wt} and BMPR2^{+/-} ECs, focusing on the differential expression of CDC42-specific regulators and effectors. In addition to BORG5, RNAseq analysis revealed notable changes in the expression levels of several CDC42 effectors in our mutant cells, notably the CDC42 effector PAK1 and the CDC42 GEFs FGD6 which have both been linked to actin cytoskeleton contractility and filopodia formation, underscoring their potential roles in tip cell dynamics. Complete RNAseq heatmap of CDC42-related genes can be found as Figure 4- figure supplement 2 with the genes corresponding to the proteins previously cited highlighted in red. These new data were discussed in our main text (lines 473-78).

Reviewer #3 (Remarks to the Author):

This is an important study by Hiepen et al. that establishes a decisive role for BMPR2 in filopodia formation during sprouting angiogenesis. Several overlapping cellular models are used to demonstrate the biochemical and biomechanical properties controlled by BMPR2 to provide directional migration especially for leading tip cells. The presented data are of high quality and rigorously analyzed. There are a few minor concerns and comments, which when addressed, will likely further strengthen the conceptual impact of the study.

1. While the requirement of BMPR2 in filopodia formation and directional migration is clear, the molecular mechanism linking BMPR2 to BORG5 and regulation of CDC42 could be further interrogated. A rescue experiment may be informative using BMPR2-kinase dead/deficient point mutants. Recent structural studies of the BMPR2 intracellular domain show that some of these point mutants associated with vascular disorders may fold and traffic relatively normally.

3.1 Answer: We thank the reviewer for their comment and we agree that investigating the role of the BMPR2 kinase function in endothelial filopodia formation is essential. To assess the importance of BMPR2 kinase activity in filopodia formation, we have compared filopodia number in BMPR2^{+/-} ECs overexpressing either wildtype BMPR2-GFP or its kinase dead mutant BMPR2-K230R in fusion with a MYC tag. Our results are presented in the new Figure 1 – Supplement 14 and show that BMPR2-K230R overexpression was not capable of rescuing filopodia formation in BMPR2-depleted cells whereas overexpression of wildtype BMPR2 could do so. These results suggest that the kinase activity of BMPR2 is essential to promote filopodia formation in endothelial cells in endothelial activation medium. We also added representative images showing typical filopodia protrusions for BMPR2^{+/-} ECs overexpressing GFP, BMPR2-GFP or BMPR2-K230R-Myc. We have addressed the subsequent findings in the main text of our manuscript (lines 218-22, 527ff).

2. It is unclear whether the BMPR2-dependent filopodia formation and sprouting are primarily occurring in vein-associated ECs (caudal vein, HUVECs)? How about the intersegmental vessels in zebrafish embryos or microvascular ECs?

3.2 Answer: Using *bmpr2b* crispant zebrafish, we detected defects in vascular sprouting of venous cells (CVP) (Figure 1a) but not in the formation of arterial intersomitic vessels (aISVs, Figure 1 – supplementary figure 5). In accordance with Wiley et al. 2011 these results suggest that in zebrafish the *Bmpr2* receptor regulates primarily venous but not arterial sprouting. Results were discussed in main text (line 158-61)

Wiley DM, Kim JD, Hao J, Hong CC, Bautch VL, Jin SW. Distinct signalling pathways regulate sprouting angiogenesis from the dorsal aorta and the axial vein. *Nat Cell Biol.* 2011;13(6):686-692. doi:10.1038/ncb2232

Minor points:

1. Fig.1b- Precisely how many embryos and ROIs are represented in the data?

3.3 Answer: In the quantification presented in figure 1b, each data point corresponds to the number of protruding cells at the caudal vein plexus of a single embryo. This information is now provided in the reporting summary and in the figure legend.

2. Please provide greater detail on how *BMPR2*^{+/-} ECs were derived. Were these CRISPR-edited cells derived from single cell clones or pooled? Compared to control, the haploinsufficient ECs appear more like a knockout.

3.4 Answer: Thank you for indicating this here. The comprehensive procedure to derive our *BMPR2*^{+/-} ECs and their detailed characterization was described in a previous publication by our group in 2019 published in PLoS Biology (Hiepen et al. 2019, PLoS Bio). For an extensive description of our *BMPR2*^{wt} and *BMPR2*^{+/-} EC models, we therefore refer to this publication in our results upon introducing these models and later in the material section as the Source and Reference for these cells.

Hiepen C, Jatzlau J, Hildebrandt S, Kampfrath B, Goktas M, Murgai A, Cuellar Camacho JL, Haag R, Ruppert C, Sengle G, Cavalcanti-Adam EA, Blank KG, Knaus P. *BMPR2* acts as a gatekeeper to protect endothelial cells from increased TGFβ responses and altered cell mechanics. PLoS Biol. 2019 Dec 11;17(12):e3000557. doi: 10.1371/journal.pbio.3000557. PMID: 31826007; PMCID: PMC6927666.

3. Co-staining with hallmark tip cell markers (e.g. CD34 or ESM1) in some of the experiments such as the leader cells in the scratch assay or spheroid sprouting would be helpful.

3.5 Answer: To further help determining the tip cell or stalk cell phenotype of ECs in our in vitro assays, we have performed CD34 staining of our cells in wound healing assays. Those resulted in ubiquitous CD34 staining in all endothelial cells, confirming the necessity for a 3D environment for CD34-specific tip cells to specify (see below). These data were therefore excluded from our manuscript. However, by performing CD34 staining in vivo in our control zebrafish and *bmpr2b* crispants, we could confirm the higher amount of CD34⁺ tip cells in control zebrafish when compared to *bmpr2b* crispants. These results are presented as new Figure 1c and discussed in text (lines 161ff).

Immunofluorescence staining of BMPR2^{wt} and BMPR2^{+/-} ECs against CD34 (green) and phalloidin (magenta). Scale bar: 20μm

New Figure list:

FIGURE 1

Figure 1a = NEW FIGURE

Figure 1b = NEW FIGURE

Figure 1c = NEW FIGURE

Figure 1d = former figure 1c

Figure 1e = former figure 1d

Figure 1f = former figure 1e

figure 1g = former figure 1f

Figure 1h = former figure 1g

Figure 1 – Figure supplement 2 = NEW FIGURE

Figure 1 – Figure supplement 3 = modified former Figure 1 – Figure supplement 2

Figure 1 – Figure supplement 4 = NEW FIGURE

Figure 1 – Figure supplement 5 = NEW FIGURE

Figure 1 – Figure supplement 6 = former Figure 1 – Figure supplement 3

Figure 1 – Figure supplement 7 = NEW FIGURE (added representative image to former Figure 1 – Figure supplement 4)

Figure 1 – Figure supplement 8 = NEW FIGURE (added quantification from former Figure 1 – Figure supplement 6 to Figure 1 – Figure supplement 5)

Figure 1 – Figure supplement 9 = NEW FIGURE (added representative pictures to Bmpr2^{wt} quantification from former Figure 1 – Figure supplement 6)

Figure 1 – Figure supplement 10 = NEW FIGURE

Figure 1 – Figure supplement 11 = NEW FIGURE

Figure 1 – Figure supplement 12 = NEW FIGURE (added representative Images to former Figure 1- Figure supplement 8)

Figure 1 – Figure supplement 13 = NEW FIGURE

Figure 1 – Figure supplement 14 = NEW FIGURE

Figure 1 – Figure supplement 15 = NEW FIGURE

Figure 1 – Figure supplement 16 = former Figure 1 – Figure supplement 9

Figure 2

Figure 2c = NEW FIGURE

Figure 2d = former Figure 2c

Figure 2e = former Figure 2d

Figure 2f = former Figure 2e

Figure 2 – Figure supplement 1 = NEW FIGURE

Figure 2 – Figure supplement 2 = NEW FIGURE

Figure 3

Figure 3e = former Figure 3 – Figure supplement 1

Figure 3f = former Figure 3e

Figure 3 – Figure supplement 1 = former Figure 3 – Figure supplement 2

Figure 4

Figure 4a = former Figure 4c

Figure 4b = former Figure 4d

Figure 4c = former Figure 4a

Figure 4d = former Figure 4 – figure supplement 1

Figure 4e = NEW FIGURE

Figure 4f = NEW FIGURE

Figure 4g = NEW FIGURE

Figure 4 – Figure supplement 1 = NEW FIGURE

Figure 4 – Figure supplement 2 = NEW FIGURE

Figure 4 – Figure supplement 3 = former Figure 4e

Figure 5 = NEW FIGURE

Deleted Figures

Former Figure 4b (was replaced by NEW FIGURE 4e)